# Warm protons at comet 67P/Churyumov-Gerasimenko – Implications for the infant bow shock

Charlotte Goetz[1,2], Herbert Gunell[3,4], Fredrik Johansson[5], Kristie LLera[6], Hans Nilsson[7], Karl-Heinz Glassmeier[2], and Matthew G. G. T. Taylor[1]

[1]European Space Research and Technology Centre, European Space Agency, Keplerlaan 1, 2201AZ Noordwijk, The Netherlands
[2]Institut für Geophysik und extraterrestrische Physik, Technische Universität Braunschweig, Mendelssohnstr. 3, 38106 Braunschweig, Germany
[3]Department of Physics, Umeå University, 901 87 Umeå, Sweden
[4]Royal Belgian Institute for Space Aeronomy (BIRA-IASB), Avenue Circulaire 3, 1180 Brussels, Belgium
[5]Institutet för Rymdfysik, Lägerhyddsvägen 1, Uppsala, Sweden
[6]Southwest Research Institute, 6220 Culebra Road, San Antonio, TX 78238-5166, USA
[7]Institutet för rymdfysik, Rymdcampus 1, Kiruna, Sweden

**Correspondence:** Charlotte Goetz (charlotte.goetz@esa.int)

**Abstract.** The plasma around comet 67P/Churyumov-Gerasimenko shows remarkable variability throughout the entire Rosetta mission. Plasma boundaries such as the diamagnetic cavity, solar wind ion cavity and infant bow shock separate regions with distinct plasma parameters from each other. Here, we focus on a particular feature in the plasma: warm, slow solar wind protons. We investigate this particular proton population further by focusing on the proton behaviour and surveying all of the Rosetta comet phase data. We find over 300 events where Rosetta transitted from a region with fast, cold protons into a region with warm, slow protons. We investigate the properties of the plasma and magnetic field at this boundary and the location where it can be found. We find that the protons are preferentially detected at intermediate gas production rates with a slight trend towards larger cometocentric distances for higher gas production rates. The events can mostly be found in the positive convective electric field hemisphere. These results agree well with simulations of the infant bow shock (IBS), an asymmetric structure in the plasma environment previously detected on only two days during the comet phase. The properties of the plasma on both sides of this structure are harder to constrain, but there is a trend towards higher electron flux, lower magnetic field, higher magnetic field power spectral density, and higher density in the region that contains the warm protons. This is in partial agreement with the previous IBS definitions, however it also indicates that the plasma and this structure are highly non-stationary. For future research, Comet Interceptor, with its multi-point measurements, can help to disentangle the spatial and temporal effects and give more clarity on the influence of changing upstream conditions on the movement of boundaries in this unusual environment.

# 1 Introduction

The plasma around comet 67P/Churyumov-Gerasimenko (67P) has been explored in depth by the instruments on board the European Space Agency's Rosetta mission (Glassmeier et al., 2007a). The Rosetta orbiter (referred to as *Rosetta* from here on) arrived at the comet in August 2014 and was passivated and impacted the surface in September 2016. The long duration of the mission enables us to explore different stages in the interaction between a comet and the solar wind. As a comet moves from aphelion to perihelion, this interaction evolves from an almost asteroid like interaction regime with very low neutral gas production to one where a fully formed bow shock and diamagnetic cavity are formed (e. g. Glassmeier, 2017; Goetz et al., 2017; Nilsson et al., 2017) along with a plethora of other identified boundaries (Mandt et al., 2016). For a more thorough review of the plasma environment of comets see Götz et al. (2019).

As a comet approaches the Sun, energy input into the surface increases which increases the amount of ice that is sublimated and escapes into space. The neutral gases, mostly water and carbondioxide, undergo photoionisation or electron impact ionisation and form a cloud of heavy ions around the comet (Hässig et al., 2015). As this cloud encounters the solar wind, the cometary ions are accelerated by the convective electric field and eventually assimilated into the solar wind flow. This process of mass-loading results in a deceleration and deflection of the solar wind in the vicinity of the comet.

Biermann et al. (1967) find that the addition of mass can be described as a source term in the mass conservation equation of a fluid description of the plasma. More recently, hybrid simulations or multi-fluid simulations have been the tool of choice to model this environment, because the large ion gyroradii of the cometary ions cannot be accurately described in a single-fluid or MHD approach. This is especially important for comet 67P, where ion gyroradii can be greater than $10000\,\mathrm{km}$ whereas the plasma cloud around the comet is restricted to radii smaller than $1000\,\mathrm{km}$ (see e.g. Koenders et al., 2016b). In this regime the cometary ions are accelerated by the convective electric field far upstream of the comet and a polarisation and ambipolar electric field closer to the nucleus (Nilsson et al., 2018; Gunell et al., 2019). The presence of the convective electric field induces an asymmetry in the interaction region that has consequences for all plasma parameters (Koenders et al., 2016a, b; Edberg et al., 2019). At higher gas production rates this asymmetry is less pronounced and the influence of the cometary ion gyroradius is diminished, because the magnetic field pile-up at the comet results in higher field magnitudes and thus smaller gyroradii.

Boundaries in the plasma at 67P have been identified and characterized in many publications. The three main boundaries that were observable by Rosetta were, in order of decreasing cometocentric distance, the solar wind ion cavity (e.g. Nilsson et al., 2017), a collisionopause (Mandt et al., 2016), and the diamagnetic cavity (e.g. Goetz et al., 2016a, b). The solar wind ion cavity is the region where no solar wind ions can be observed in the plasma, from May 2015 to January 2016 Rosetta was almost exclusively within this region. The collisionopause demarcates the tenuous boundary where ion-neutral or electron-neutral collisions become important and it has been shown to lie within the solar wind ion cavity. Finally the diamagnetic cavity is the innermost observed region, where the magnetic field is very close to zero. For a more detailed overview of these boundaries see e.g. Götz et al. (2019).

Another boundary in the plasma environment of a comet, but not observed by Rosetta, is the bow shock. In the classical fluid description by Biermann et al. (1967) and Flammer and Mendis (1991), the mass-loading of the solar wind flow results in a deceleration until a critical point is reached and no mass can be added. A cometary number density of just a few percent is sufficient here, so the critical point is reached already far upstream of the nucleus. There, the interaction between the solar wind and the comet cannot be described by mass-loading alone, instead the flow changes from supersonic to subsonic and a bow shock forms. This prediction is shown to fit well with observations at e. g. comet Halley (Neubauer et al., 1986), where the bow shock was detected $1.15 \times 10^6$ km from the nucleus. The transition from unshocked to shocked solar wind was identified by a decrease in speed, increase in density and temperature and an increase in the magnetic field (Coates et al., 1990). The shock was identified as a low Mach number shock, in agreement with the model, which predicted a gradual slowing of the solar wind flow already upstream of the shock due to the incorporation of the cometary ions. The cometary ion density is often neglected in bow shock models at high activity comets, because it only reaches 1.5-2.5% of the total density. Observations of bow shocks at other comets where quite similar, although at the lower activity comets Giacobini-Zinner (GZ) and Grigg-Skjellerup (GS) the bow shock is often termed a bow wave, due to the lack of a sharp boundary (Smith et al., 1986). At GS, a strong non-gyrotropy of the cometary ions could be observed near the bow wave, together with wave activity triggered by this unstable distribution function (Coates et al., 1996). Koenders et al. (2013) compare the bow shock distances from a simple single-fluid model with distances gained from Hybrid simulations and find that the fluid models predicted consistently higher stand off distances. Thus, the ion gyroradius effects are pronounced even in the most fluid-like stage of the plasma around comet 67P.

The shock itself forms by waves steepening into the nonlinear regime. The speed of the steepened wave is faster than that of the linear wave, but steepening is counteracted by dissipation. If an obstacle and a plasma are in relative motion faster than the speed of linear waves, the waves steepen until an equilibrium is reached where the shock becomes a stationary wave in the obstacle's (the comet's) frame of reference (Balogh and Treumann, 2013).

Bow shocks have been studied at comets (Simon Wedlund et al., 2017) and elsewhere in the solar system (see e. g. Martinecz et al., 2008; Fahr and Siewert, 2015; Hall et al., 2016), but so far the development of a bow shock could not be observed, simply because a bow shock had already been fully formed. Comets provide an excellent laboratory to investigate the process of bow shock formation, where the gradual increase in gas production rate over weeks or months means that the intermediate stages of this interaction can be observed and studied.

The pre-cursor of a bow shock, the infant bow shock (IBS), was first first reported by Gunell et al. (2018). They show two cases from different days of fast changes in the cometary plasma and associate these changes to an asymmetric structure in the solar wind flow that is also found in Hybrid simulations. To detect the IBS the authors look for a change in the magnetic field direction. This ensures that the IBS moves over the spacecraft fairly quickly and the boundary is clearly detectable. This is necessary as Rosetta has a negligible velocity with respect to the boundary. From two days of data it is found that the magnetic field reversal was coincident with an increase in magnitude and wave activity. At the same time the proton velocity distribution function becomes broader and the bulk velocity decreases. The electron flux and energy increases. These observations are compared with hybrid simulations of the comet at similar gas production rates. While the reversal of the magnetic field is used to ensure a fast transition of the bow shock from one side of the comet to the other, the main signature of the infant bow

shock is the presence of warm, slow protons. According to Balogh and Treumann (2013) , the slowing down and heating of the medium over a narrow layer or boundary is the defining feature of any shock. A highly asymmetric boundary is seen in the simulations and the simulated proton spectra are very similar to what is observed by Rosetta. Thus the observations are found to be consistent with the detection of an asymmetric boundary. As this boundary is similar to a bow shock at a fully developed comet, it is termed the infant bow shock. Prior to the first identification of an IBS, it was observed that similar signatures seen earlier in the mission could possibly be the result of "the crossing of a plasma boundary" (Edberg et al., 2016).

This work aims to study the warm proton signatures first associated with this infant bow shock, but with a broader scope. We then characterize the plasma changes at the boundary as well as its location and discuss how these signatures are related to the infant bow shock and its characteristics.

## 2   Data

### 2.1   Instruments

For this study we use all sensors of the Rosetta Plasma Consortium (Carr et al., 2007) as well as the neutral gas monitor ROSINA-COPS (Balsiger et al., 2007), which provides the neutral gas density as context for the plasma measurements.

The Ion Composition Analyzer (ICA, Nilsson et al., 2007) can provide mass separated differential energy flux with a temporal resolution of $192\,\mathrm{s}$. The field of view (FoV) is $90°$ in elevation and $360°$ in azimuth, but there are some obstructions from the spacecraft. Thus, the solar wind signal is not always detected even when it is present in the plasma. Especially rotations of the spacecraft and of the magnetic field can lead to a loss of the solar wind signature. Such loss in the signature is typically seen as a drastic reduction in the solar wind flux/density. Often, the signal is then still visible in the RPC-IES instrument, as the FOV is partially complimentary (rotated by $60°$), a detailed description of the FoV can be found in the ICA User Guide on the PSA[1]. Solar wind densities near the comet also decrease due to significant charge exchange losses (Simon Wedlund et al., 2019). This caused rather low densities in the times when Rosetta was just outside the solar wind ion cavity. The RPC-ICA solar wind moments, including the temperature, used in this study are integrations of the RPC-ICA PSA L4 PHYS-MASS data set, also delivered to the planetary science archive (PSA) as RPC-ICA L5 MOMENT data set.

The Ion and Electron Sensor (IES, Burch et al., 2007) provides differential energy flux for electrons and ions (without mass resolution). The time resolution is at least $256\,\mathrm{s}$, and the measurements at low energies are disturbed by the spacecraft potential, which is between $0\,\mathrm{V}$ and $-20\,\mathrm{V}$ most of the time. For a quantitative analysis of the electron flux, we use the method detailed in Lavraud and Larson (2016) to correct the fluxes and energy bins for the spacecraft potential. We also calculate the flux at $60\,\mathrm{eV}$, and $120\,\mathrm{eV}$ as a time series for the statistical study.

The LAngmuir Probe instrument (LAP, Eriksson et al., 2007) and the Mutual Impedance Probe (MIP, Trotignon et al., 2007) are used to provide plasma density estimates (see e.g. Breuillard et al., 2019; Johansson et al., 2020) and measurements of the spacecraft potential (Odelstad et al., 2015). Although the absolute uncertainty of each individual LAP measurement of

---

[1]https://cosmos.esa.int/web/psa/rosetta

spacecraft potential may become large (typically 30%) the random noise of the spacecraft potential and the cross-calibrated LAP-MIP density dataset is believed to be only a few percent, and as such suitable for statistical studies (Johansson et al., 2020). For this study, we use the density estimate to characterize the plasma.

The magnetic field measurements are provided by the MAGnetometer (MAG, Glassmeier et al., 2007b). Although the maximum sampling rate of the instrument is 20 Hz, only resampled data with 1 Hz sampling rate is used here, which is sufficient for the study of large scale structures. The magnetic field can only be determined within an accuracy of $\sim 3\,\mathrm{nT}$ per component, due to the influences of spacecraft fields and sensor temperature variations (Goetz et al., 2016a, 2017).

## 2.2    Selection of Intervals

In this study we investigate where and when warm protons are detected near the comet. We aim to identify the regions with warm protons and the boundary that separates the warm and colder proton populations and how the plasma properties react to this boundary. Since we are interested in intervals where Rosetta can still measure the solar wind ions, we do not examine days in which Rosetta was in the solar wind ion cavity (Nilsson et al., 2017), i.e. approximately six months around perihelion (August 2015). All days with characteristics similar to those shown in Gunell et al. (2018) are then flagged. We find 152 days
with detections.

To constrain the regions with warm, slow plasma, we then inspect every pre-selected day and set start and end times for each interval in which warm, light ions are detected. This is done according to the following criteria: the solar wind ions measured by ICA and/or IES need to be at significantly lower energies (smaller mean speeds) and show a broader (higher temperature) distribution as compared to surrounding intervals. For the first criterion, the threshold was a shift of the peak of the ion spectra by at least three energy bins, corresponding to at least 60 eV. We only use these two criteria for detection. For verification we
evaluate additional properties like the ICA derived proton temperature, plasma density, suprathermal electron fluxes, magnetic field magnitude power spectral density in the frequency range between 50 mHz and 75 mHz and the magnetic field magnitude. However, the direction of change (increase or decrease) is not considered, because the change in parameters is simply an indicator that the change in proton energy and flux is not due to instrumental or spacecraft effects. The measurements of the
electron energy are changing over time as one half of the detector decreases in sensitivity. Thus even small changes in spacecraft attitude or magnetic field direction can have significant consequences in the electron countrate. An additional parameter for characterization is the angle $\theta$, defined by its cosine:

$$\cos\left(\theta\right) = \frac{\boldsymbol{E} \cdot \boldsymbol{x}}{|\boldsymbol{E}| \cdot |\boldsymbol{x}|} = \frac{(-\boldsymbol{v} \times \boldsymbol{B}) \cdot \boldsymbol{x}}{|\boldsymbol{v} \times \boldsymbol{B}| \cdot |\boldsymbol{x}|} \tag{1}$$

where $\boldsymbol{x}$ is the spacecraft position in CSEQ, $\boldsymbol{v}$ is the velocity of the undisturbed solar wind and $\boldsymbol{B}$ is the magnetic field.
The angle $\theta$ was introduced by Gunell et al. (2018) to facilitate a comparison between observations and simulations such as those conducted by Lindkvist et al. (2018). A positive value of the observational $\cos\left(\theta\right)$ corresponds to a location in the $+E_c$ hemisphere of simulation space, that is to say, where the coordinate pointing in the same direction as the convective electric field of the solar wind is positive. We also use the sun aspect angles of the spacecraft to exclude an attitude change of the

| Start time | H$^+$ E/q | $\Gamma_{IES,e}$ | $B_m$ | $P_B$ | $\cos(\theta)$ | $n_{pl}$ | $T_p$ | H$^+$ E/q | $\Gamma_{IES,e}$ | $B_m$ | $P_B$ | $\cos(\theta)$ | $n_{pl}$ | $T_p$ |
|---|---|---|---|---|---|---|---|---|---|---|---|---|---|---|
| Dec 07, 14 03:49 | ↓ | ↑ | – | ↓ | – | – | ↑ | ↑ | – | – | – | – | – | – |
| Dec 25, 14 09:50 | ↓ | ↑ | ↓ | ↓ | – |  | ↑ | ↑ | ↓ | – | – | – |  | ↓ |
| Jan 04, 15 12:19 | ↓ | ↑ | – | ↓ | – |  | ↑ | ↑ | – | – | ↑ | – |  | ↓ |
| Jan 04, 15 19:55 | ↓ | ↑ | ↑ | – | – |  | – | ↑ | ↓ | ↑ | – | ↓ | ↑ |  |
| Mar 07, 15 05:48 | ↓ | ↑ | ↑ | ↑ | ↑ |  | ↓ | ↑ | ↓ | ↓ | – | ↓ | – |  |
| Feb 10, 16 09:02 |  |  |  |  |  |  | ↑ | ↓ | ↓ | ↓ | ↓ | ↑ | ↑ |
| Feb 26, 16 05:50 | ↓ | – | – | – | – | – | – | ↑ | – | ↓ | – | ↓ | – | – |
| Feb 29, 16 00:27 | ↓ | – | – | – | ↓ | – | – | ↑ | ↓ | ↓ | ↓ | ↑ | – | ↑ |
| Apr 08, 16 03:27 | ↓ | ↑ | ↓ | ↓ | ↑ | ↑ |  | ↑ | ↓ | ↑ | ↑ | ↓ | ↓ |  |
| Apr 08, 16 07:58 | ↓ | ↑ | ↓ | ↑ | – | ↑ | – | ↑ | ↓ | ↑ | – | ↓ | ↓ | – |
| Jun 01, 16 12:11 | ↓ | ↑ | ↑ | ↑ | ↑ | ↑ | ↑ |  |  |  |  |  |  |  |
| Jul 09, 16 12:43 | ↓ | ↑ | ↑ | ↑ | – | ↑ | ↑ | ↑ | ↓ | – | – | ↓ | – | ↓ |
| Jul 09, 16 15:52 | ↓ | ↑ | ↓ | ↓ | – | ↑ | ↑ | ↑ | ↓ | – | – | ↑ | ↓ | ↓ |
| Median | ↓ | ↑ | – | – | – | –↑ | ↑ | ↑ | ↓ | – | – | ↓ | – | ↓ |

**Table 1.** List of 13 events chosen for a more detailed study and list of parameter changes when crossing from upstream to downstream (inward, left) and from downstream to upstream (outward, right). The last line summarizes events by giving a median change. Missing signs indicate that no data was available.

spacecraft as a reason for a change in the proton signal. These are defined as the angles of the three spacecraft axes to the
Sun-comet line[2]. Events that coincide with major attitude changes ($> 10°$) are not included in the study.

This selection is slightly different than the criteria used in Gunell et al. (2018), because we would like to investigate the occurrence of the warm protons regardless of the direction and variability of the magnetic field. Other parameter changes like solar wind velocity and density as well as cometary ion density can also move the boundary, causing warm protons to appear at the spacecraft (as stated in Gunell et al., 2018). Thus, limiting the dataset to cases with magnetic field reversals would
unnecessarily constrain the number of events.

Event selection was carried out manually rather than automated. Due to the complicated nature of instrument operations by Rosetta, one cannot expect a consistent set of data for all events that could be used in a detection algorithm. For example, a slew of the spacecraft may have rotated the solar wind ions out of the FoV of ICA, meaning that we would not be able to characterize this interval. However, visual inspection quickly shows that the solar wind ions are now in the FoV of IES,
allowing for a characterization of the event. No combined dataset exists so far. Thus a selection by hand was determined to be the best course of action. For replicability, the list of events with start and end times is given in the supplementary online material.

[2]See https://www.cosmos.esa.int/web/spice/spice-for-rosetta

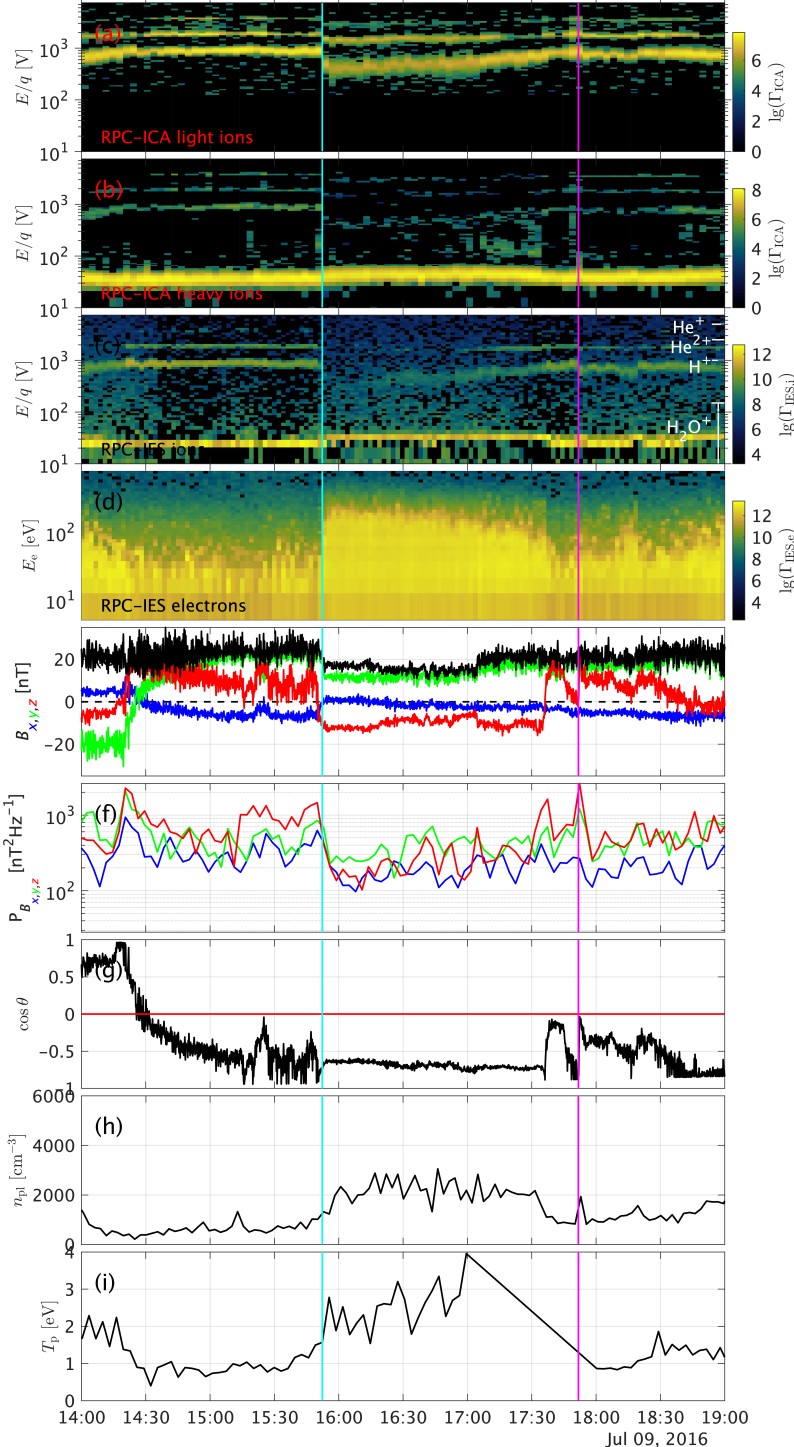

**Figure 1.** Observations of the event on July 9th, 2016. From top to bottom: a) ICA solar wind ions, b) ICA heavy ions, c) IES ions, d) IES electrons, e) magnetic field in CSEQ coordinates, f) magnetic power spectral density in the frequency range between 2 mHz and 15 mHz, g) angle between spacecraft position and convective electric field, h) plasma density from LAP, and i) 1D proton temperature from ICA.

## 3 Analysis

From the observations, we identify a total of 370 events where Rosetta observed warm protons. In analogy to the simulation we will refer to the part of the plasma that is warm as the downstream (index $d$) part of the plasma, while the plasma with more pristine solar wind is referred to as the upstream (index $u$) region. In some cases, the identification of the downstream plasma is impossible due to data gaps. In this case, the event ends (or begins) at the data gap. As the particle spectra are especially difficult to condense into simple scalar parameters that are statistically representative of a large dataset, we start by examining a subset of 13 events in Sect. 3.1. This is complemented by a statistical treatment of some quantities derived from the whole dataset in Sect. 3.2.

### 3.1 Detailed investigation of a small subset of events

We begin with a detailed investigation of a smaller subset of events. These events were chosen somewhat arbitrarily, so as to represent a broad picture of the situation, however emphasis was put on events that were easily visible for illustration purposes. Thus, we have chosen 13 events, as listed in Table 1.

Fig. 1 shows one example event, observations for the other events can be found in the appendix (Fig. A1 and A2). The blue line indicates the time when Rosetta passed from the upstream to the downstream region, the magenta line shows the outbound pass. From top to bottom: we see a pronounced decrease in the proton energy as well as a broadening of the energy band in the downstream region. This was the criterion of selection for the events. The energy of the heavy ions (panel b) is increased slightly. The IES signal of the protons (panel c) is lost at first, but then the protons return to the field of view and appear broader and slower. The $He^+$ and $He^{2+}$ show a very similar behaviour to the protons (panel a), decreasing and broadening in energy, but their signature remains distinct from each other at all times. The IES electron signature (panel d) increases in energy and flux. Interestingly, the flux diminishes at the same time that the proton energy increases gradually, implying that the spacecraft moved slowly upstream in a shock-fixed frame of reference into a region with less electron heating and a less slowed-down proton distribution. This is similar to what was observed already by Gunell et al. (2018). The magnetic field (panel e) decreases in magnitude and the z-component changes sign. The power spectral density of the magnetic field (panel f) is decreased. The angle $\theta$ does not change significantly (panel g), but the magnetic field direction does change. This is because $\theta$ represents the angle between the x-axis and electric field, thus it does not reflect changes in the z-component of the magnetic field very well. The plasma density (panel h) as well as the proton temperature (panel i) are higher in the downstream region. For this initial study we have categorized the change at the two boundary regions for all 13 events. Since no data is available for the outbound pass for one event and for the inbound pass for one event, we have a total of 12 events for each the inbound and outbound pass. The parameters that we use to characterize how the plasma changes at the boundary are the proton energy $H^+$ E/q, the flux of the electrons $\Gamma_{IES,e}$, the magnetic field magnitude $B_m$, the power spectral density of the magnetic field $P_B$, the angle $\cos(\theta)$, the plasma density $n_{pl}$, and the proton temperature $T_p$. The changes are indicated in Table 1. Here, we are only looking at the qualitative changes, quantitative changes will be assessed in the next section, where the larger statistics should make up for the large uncertainty for each event. These clear, qualitative events can then be used to verify the quantitative, statistical outcome.

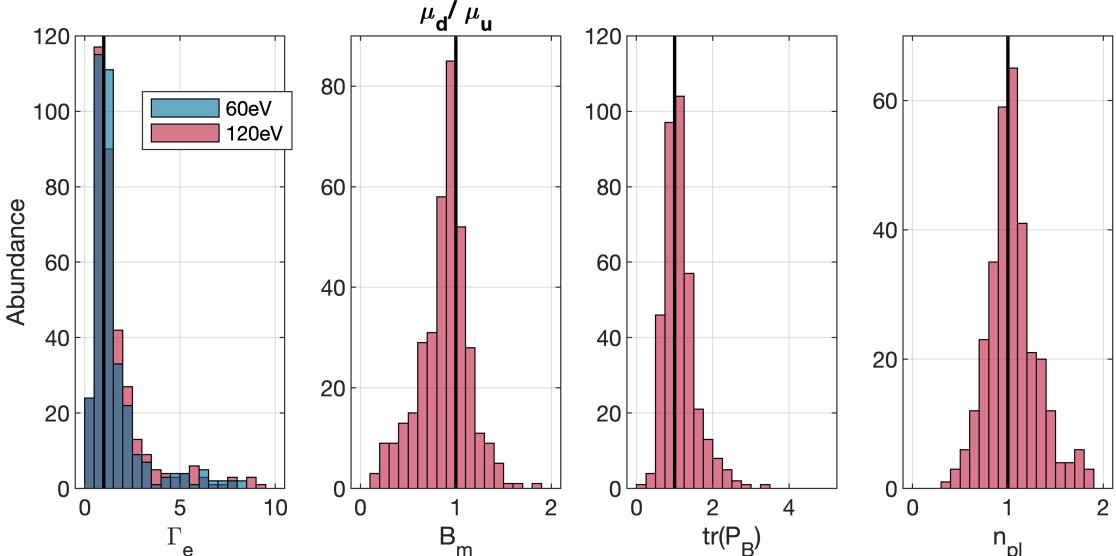

**Figure 2.** Comparison of the upstream and downstream mean values for four of the seven parameters chosen for investigation. From left to right: Electron flux $\Gamma_e$ at $60\,\mathrm{eV}$ (blue) and at $120\,\mathrm{eV}$ (red), magnetic field strength $B_m$, trace of the magnetic field power spectral density $\mathrm{tr}(P_B)$, and plasma density $n_{pl}$.

From these events we can conclude: Since the proton energy was used as a selection criterion the proton energy in the downstream region is always lower than upstream. The proton temperature is almost always higher in the downstream region. For the other parameters, we find that the energy of the electrons is almost always increased and the density is often higher in the downstream region. For the other parameters, no such clear pattern emerges. The magnetic field and magnetic field power

spectral density are sometimes increased, sometimes decreased, with no apparent pattern. The angle of the field can change, but events without field changes also exist. It is also interesting to note that not all events have a sharp boundary. For example, the event on Feb 10th, 2016 (Fig. A1, bottom right) shows that the transition can sometimes be very broad. We see that it takes about 20 minutes for the magnetic field to change direction, and for the proton flux to gradually increase in width and decrease in energy. The beginning and end of this transition period are marked with two lines, both magenta in colour.

## 3.2 A statistical study of the entire event dataset

Here, we expand the discussion to the entire dataset of 370 events.

We investigate the behaviour of the same parameters as above, but now for all events. The statistical assessment of the proton flux is complicated by an incomplete FoV and the broad distribution of the protons. Therefore, moments of the distribution function are less representative in the situation at comet 67P. Therefore a direct statistical study of the moments cannot be

conducted. To assess the electron flux changes, we chose two energy values ($60\,\mathrm{eV}$ and $120\,\mathrm{eV}$) to extract a 1D time series of the flux at these energies. They were chosen based on an inspection of the subset of events, where these energy bands showed

the clearest change. But we should also bear in mind that the IES electron detector decreased in sensitivity in the latter half of 2015 (Madanian et al., 2020), thus electron fluxes may be below the noise level quite often, especially in 2016. We have not included the angle $\theta$ in this investigation, because the angle of the field is very susceptible to magnetometer offset problems, especially for low field regimes and requires a visual inspection that is not possible for a statistical study.

Figure 2 shows histograms of the ratio of the downstream to the upstream mean parameters $\mu_d/\mu_u$. To calculate this ratio we use an interval of 18 minutes before and after the event, as well as all the observations of the downstream plasma. The interval length was also varied from 10 minutes up to an hour, but the overall results were not impacted by that.

These larger statistics agree mostly with the observations from the 13 events that were categorized by hand. From left to right:

$\Gamma_e$ In our smaller subset, the energy of the electrons in the $60\,\mathrm{eV}$ and $120\,\mathrm{eV}$ band increases in 10 of the 12 inbound passes and decreases in 8 of the 12 outbound passes. In the entire dataset the electron energy is increased in the downstream region in $60\%$ of all cases. That the larger statistics do not show the same behaviour may in part be because the energy dependent electron flux is difficult to condense to a single parameter, and the instrument sensitivity declined significantly after perihelion. We have observed cases where the flux was very low and thus changes were not visible.

$B_m$ The magnetic field decreases in $68\%$ of cases. This is consistent with the case studies above.

$\mathrm{tr}(P_B)$ The trace power spectral density increases downstream in $58\%$ of all cases.

$n_{pl}$ The plasma density increases in $52\%$ of all cases. This is consistent with the case studies, where the density was either increased downstream or not changed at all.

## 3.3 Location of the events

From Gunell et al. (2018) the expectation is that the IBS and the associated warm ions are detected preferentially at the intermediate activity stage of comet 67P. To investigate this, we estimate the gas production rate $Q$ from the in-situ neutral gas density measurements under the assumption of a simple, spherical outgassing model (Haser, 1957).

Figure 3 shows the occurrence location of the events dependent on the outgassing rate and heliocentric distance. To check for observation bias, the trajectory of the spacecraft is indicated in grey. We find that most detections are made between $Q = 10^{26}\,\mathrm{s}^{-1}$ and $Q = 6 \times 10^{27}\,\mathrm{s}^{-1}$, which corresponds roughly to heliocentric distances between $1.7\,\mathrm{AU}$ and $2.7\,\mathrm{AU}$. The cometocentric distance of the events increases with increasing gas production rate.

The convective electric field $E_c$ in the solar wind accelerates the cometary ions in only one direction and, to conserve momentum, the solar wind is deflected in the opposite direction (Coates et al., 2015; Deca et al., 2017). We transform the position of the events that we identified into a Cometocentric Solar Electric field (CSE) system. Specifically, the z-axis is aligned with $-\boldsymbol{v_{sw}} \times \boldsymbol{B}$, where $\boldsymbol{B}$ is the measured magnetic field at the spacecraft location and $\boldsymbol{v_{sw}}$ is the solar wind velocity. For this comparison, the solar wind direction is simply assumed as the direction away from the Sun. This has been shown to be a good estimator for the CSE system (Edberg et al., 2019). The local solar wind velocity at this stage in the cometary

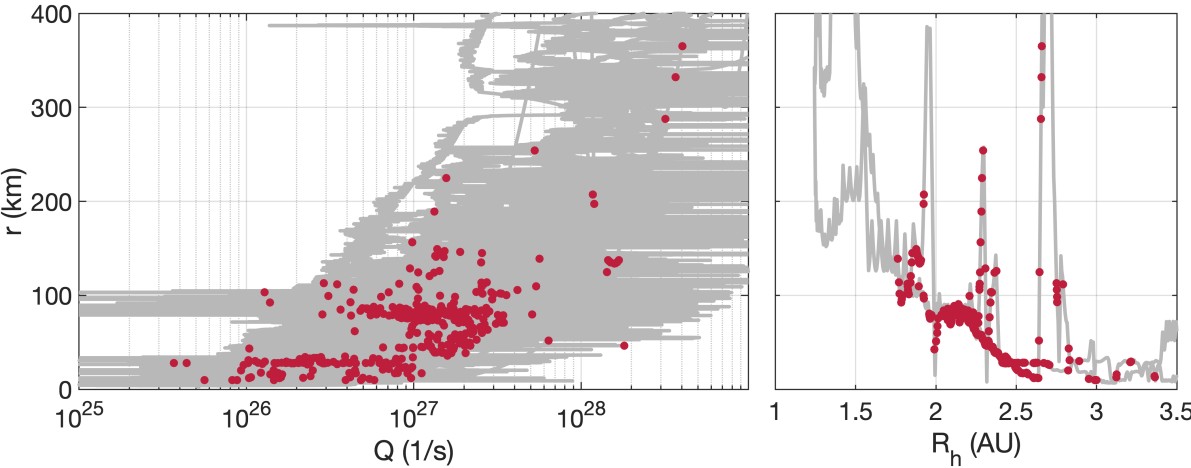

**Figure 3.** Cometocentric distance of the spacecraft over gas production rate (left) and heliocentric distance (right). The gas production rate was derived from measured neutral gas densities using a spherically symmetric model. The grey lines show the position during the entire Rosetta mission, while the red dots indicate boundary crossings.

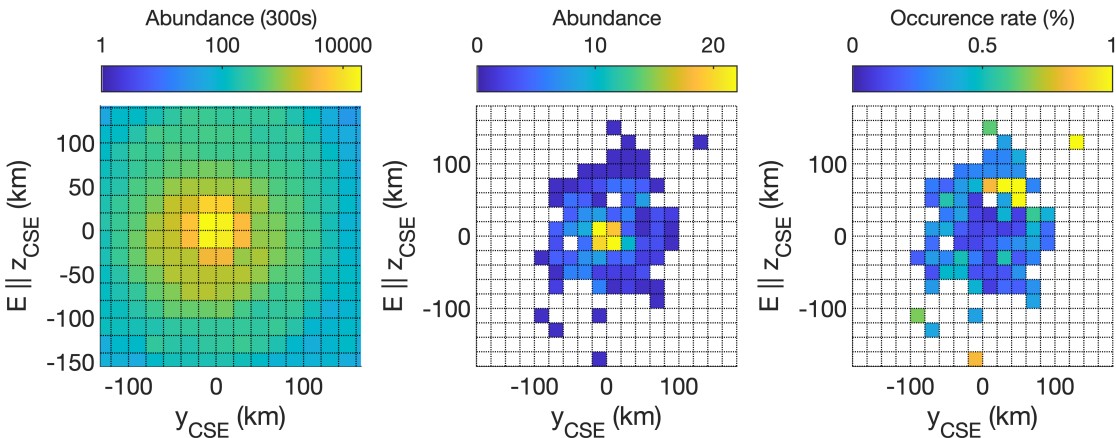

**Figure 4.** Abundance of the position of the spacecraft (left), position at which warm protons were detected (middle) and occurrence rate of detections normalized to the spacecraft dwell time (right). The $+E_c$ hemisphere is that of $z_{CSE} > 0$.

development is not representative of the upstream solar wind direction, because of deflection (Nilsson et al., 2017). This can also be seen in simulations (e. g. Fig. 5), where the deflection of the solar wind ions becomes significant very close to the infant bow shock, exactly in the region that is used for calculation of the electric field direction. Thus a more accurate estimate is impossible as undisturbed solar wind observations are not available at the comet.

The left panel of Fig. 4 shows the location of the spacecraft in this CSE system binned in 300s intervals. There is very good coverage of the entire coma in the terminator plane. We chose to limit our investigation to the $y_{CSE} - z_{CSE}$-plane, because Rosetta was for the most time very close to the nucleus in a terminator ($x = 0$) orbit and coverage in the $x$-direction is insufficient. In the middle panel, the location of the events is shown in the same coordinate system. The right panel shows the occurrence rate of warm proton detections. The most detections are in the $(+y,+E_c)$ quadrant, at about $70\,\mathrm{km}$ from the nucleus.

## 4 Discussion

We have performed a statistical study of periods where Rosetta observed protons with higher temperature and lower mean energy than the solar wind. It is worthwhile to discuss whether all or only a subset of these were observations of an infant bow shock as reported by Gunell et al. (2018).

On average the flux of electrons does increase downstream of the structure, which is in agreement with the properties of the IBS as reported in the previous study. However many events also show an opposite behaviour. The agreement is better in the smaller dataset of 13 cases than in the complete dataset. Thus, at least some of this discrepancy might be attributable to the inability of the flux at $60\,\mathrm{eV}$ or $120\,\mathrm{eV}$ to accurately represent the electron spectra due to FOV and spacecraft charging effects (see Sect. 2.1).

For the power spectral density of the magnetic field one would, for a shock, expect an increase downstream of the shock where oscillations are known to occur. We see this in 58% of the cases of the whole data set, but only in 32% of the cases in the smaller subset. Although events without an enhanced suprathermal electron flux or increased magnetic power spectral density could be examples of phenomena other than an IBS, it is not necessarily so. The region downstream of a shock is structured. Oscillations are expected to peak close to the shock and decay farther downstream (Ziegler and Schindler, 1988), and also the scale lengths are different for electrons, protons and heavy ions. It is thus possible for the spacecraft to probe only a region with hot protons without passing through regions with hot electrons or large amplitude wave activity. This would mean that the spacecraft probed parts of the equivalent to a planetary magnetosheath, whose properties depend on the presence of a bow shock, but without crossing the bow shock itself. This must at times occur, given the slow motion of the spacecraft of the order of $1\,\mathrm{m\,s^{-1}}$.

As the spacecraft moves very slowly the observations rely on changes in the upstream conditions for the spacecraft to pass from one region to another. Gunell et al. (2018) only showed events where the location of the spacecraft in a convective electric field aligned system changed, which is caused by a change of the direction of the interplanetary magnetic field. In this study, we do not rely on the convective electric field to change (represented by the angle $\theta$). Any change in upstream conditions

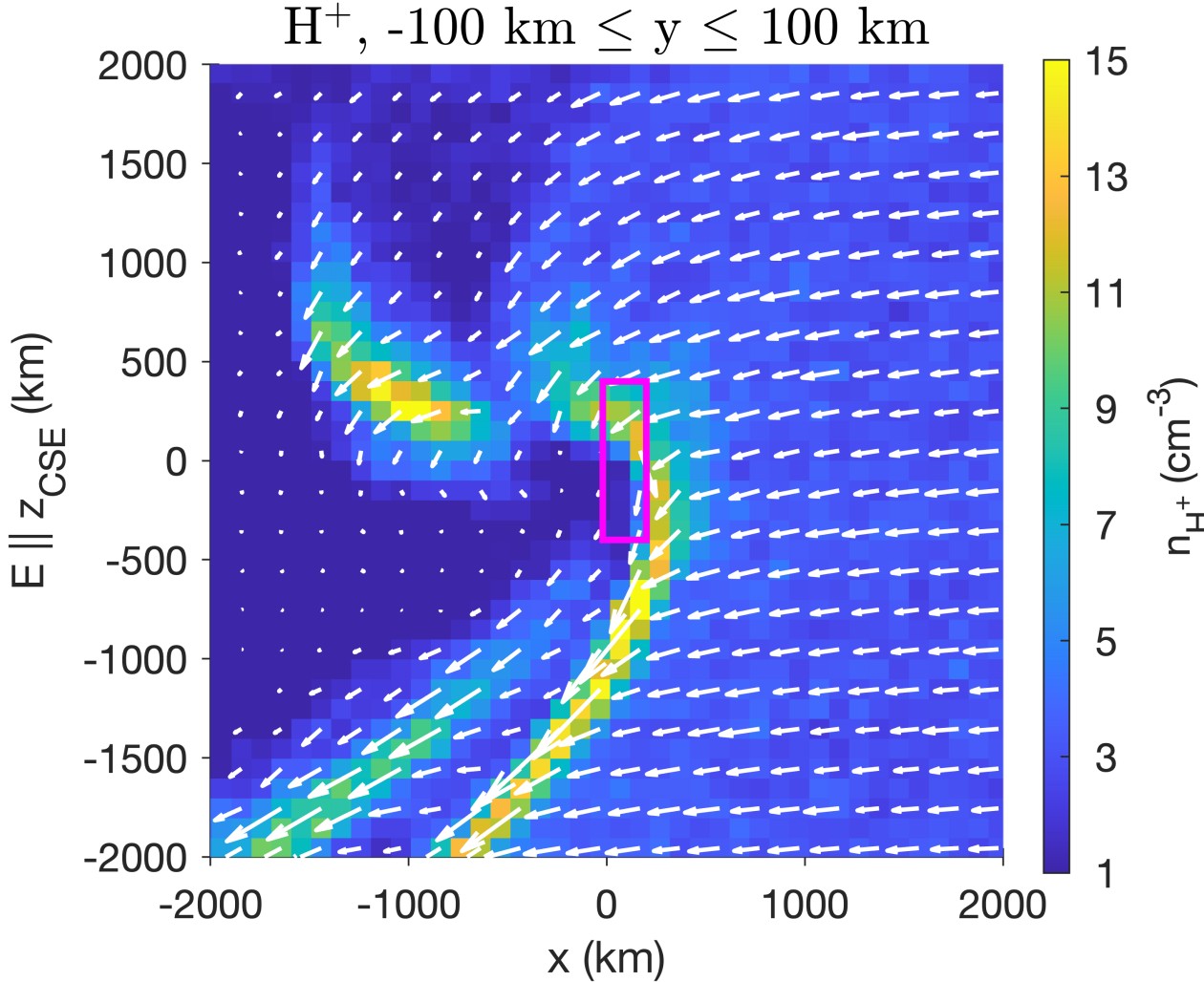

**Figure 5.** Density and direction of the flux of the protons from the Hybrid simulations. The simulation was run for a case of $Q = 3.2 \times 10^{27}\,\mathrm{s}^{-1}$. For a more detailed list of parameters see Gunell et al. (2018). Here, the Sun is to the right. The IBS is roughly located where the proton density reaches its highest values (yellow).

that moves the boundary can also allow Rosetta to cross it. This gives us access to a larger set of events, but we do not know what upstream condition change leads to the boundary movement. Our observations will therefore be affected not only by the change of one specific solar wind parameter, but also by the correlation between the different solar wind parameters that change together systematically. We have also shown that some events have very broad transitions regions (10s of minutes), a scale over which the plasma has previously been shown to be extremely variable. The relevant gyroperiods of $0.5\,\text{s}$ (protons) and $9\,\text{s}$ (water ions) are much smaller than any of the transition times we observe. The behaviour of the magnetic field magnitude is an example of this. In shock modelling, the magnetic field is generally stronger on the downstream than the upstream side of the shock. In our statistics, we have many cases of the opposite behaviour. One possibility is that an increase in the solar wind dynamic pressure increases the mass-loading threshold of the plasma (Biermann et al., 1967) which means that the critical condition for a shock is met later in the flow, and thus closer to the comet. This moves the IBS further towards the nucleus and Rosetta passes into the upstream region, but at the same time the magnitude of the interplanetary field increases, resulting in a new, stronger magnetic field. Evidence for such a correlation in the solar wind has been found before (see Fig. 4 of Maggiolo et al., 2017). To separate the behaviour of the solar wind itself from the cometary response to the solar wind one would need an upstream monitor in the vicinity of the comet. Thus, the observations are not inconsistent with the theory of an infant bow shock.

We can also consider just the subset of events where the plasma behaves as expected for an IBS (the magnetic field increases downstream along with an increase in the power spectral density, increase in electron flux). In about $10\,\%$ of the cases all parameters that were evaluated, the magnetic field included, behave as expected at the same time. One such event is shown in Fig. 6. Although the ICA data is missing for the first half of the event (before 06:30), we can clearly see warm proton fluxes in the IES data while ICA is off. Once ICA is running, the protons do appear in the ICA energy spectra. The electron flux, magnetic field magnitude and power spectral density are increased in the region with warm protons. This event is very similar to that shown in Gunell et al. (2018). We therefore conclude that these $10\%$ are definitely the same IBS structure as reported before. For the events that do not comply with these criteria, we have examined if they only occur under specific circumstances (i.e. position, gas production rate, spacecraft potential), but no apparent pattern emerges. There are also no correlations between the changes in field and changes in the electron flux. For the events where the magnetic field increases, the flux of electrons can both increase and decrease. The same goes for the plasma density, as well as the power spectral density.

We present here also for the first time the plasma density measurements for this boundary. We find that the density of the plasma on average does not change significantly at the boundary. In fact, events where the plasma density increases, decreases and is unchanged can all be found in the data set. This was expected, as the plasma density at 67P at this point is dominated by the heavy ions and not the solar wind. We can estimate the fraction of cometary ions for the event shown in Fig. 1. The cometary ion density is of the order of $1000\,\text{cm}^{-3}$ and we can estimate the maximum proton density from a simple back-of-the-envelope calculation: assuming a solar wind density of $3\,\text{cm}^{-3}$ (typical for heliocentric distances around 2 AU) and a compression factor of $\sim 4$, we get a proton density of $12\,\text{cm}^{-3}$. This is close to what is also observed in the simulation used below. This gives a fraction of $\sim 99\%$ cometary ions. Even if this estimate is very rough, it is clear that the cometary ions are at this point clearly

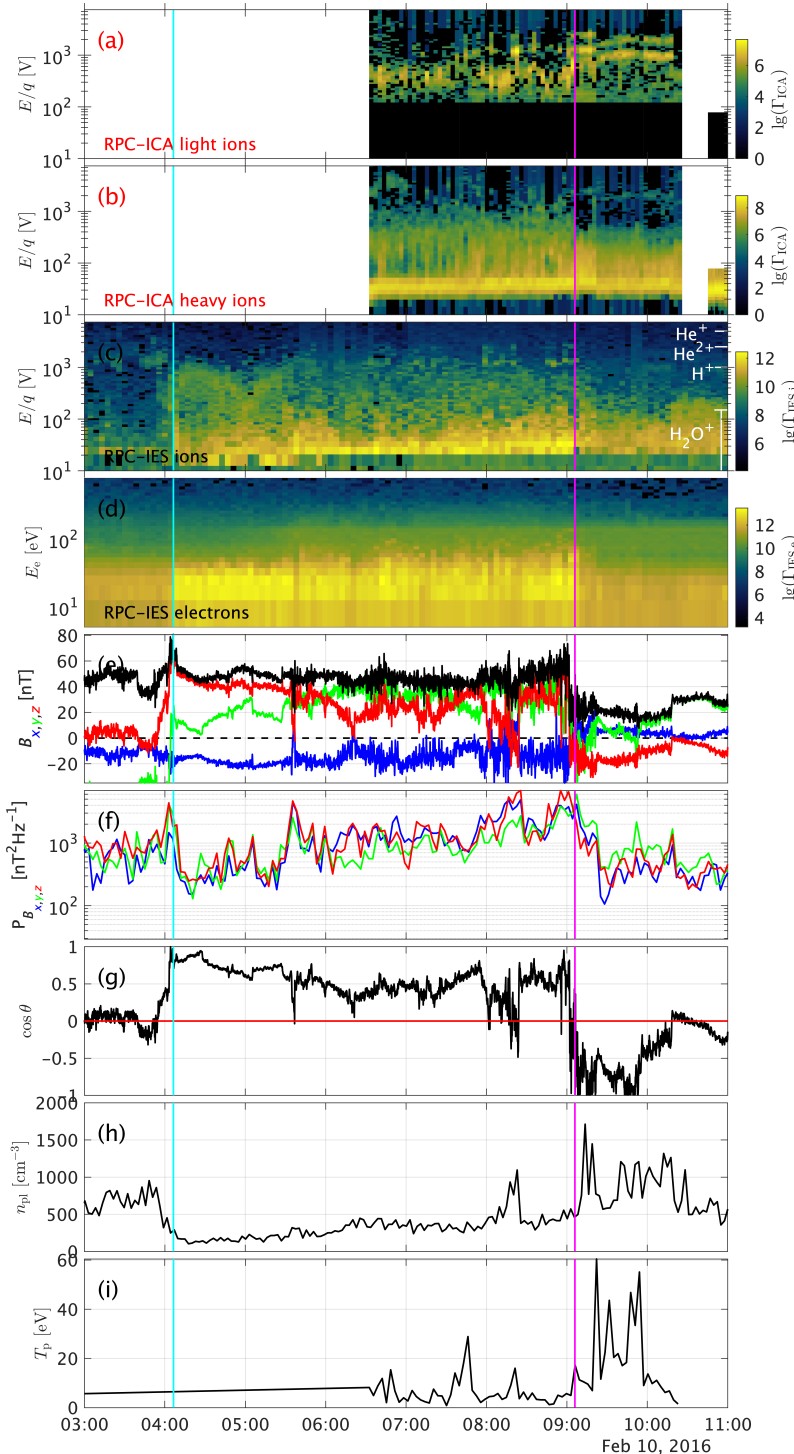

**Figure 6.** Observations of the event on February 10th, 2016. From top to bottom: a) ICA solar wind ions, b) ICA heavy ions, c) IES ions, d) IES electrons, e) magnetic field in CSEQ coordinates, f) magnetic power spectral density in the frequency range between $2\,\mathrm{mHz}$ and $15\,\mathrm{mHz}$, g) angle between spacecraft position and convective electric field, and h) spacecraft attitude.

dominating the plasma density and the solar wind has only very little influence density-wise. Instead Williamson et al. (2020) found that the solar wind and cometary ion momentum are of similar importance at the intermediate stage of cometary activity.

From the observations, we learn that the warm protons are mostly found in the $(+y,+E_c)$ quadrant in a solar wind convective electric field frame. We can compare this to Hybrid simulations of the plasma environment at similar gas production rates (Lindkvist et al., 2018). Fig. 5 shows the proton density and flux (arrows) in a hybrid simulation. For a detailed description of the simulations and the input conditions used see Lindkvist et al. (2018) and Gunell et al. (2018). The infant bow shock as identified in the previous study is visible as the large asymmetric region of enhanced proton density. The purple box shows the approximate region that Rosetta was able to measure in. The simulation reveals a more detailed structure than a simple test particle picture. For example, the proton density enhancement in the upper left corner as well as the secondary IBS structure are due to the gyration of the protons. As the protons are accelerated by the electric field their trajectories form cycloids. In these two regions many of these cycloidal trajectories reach their cusps where the velocity is low, thus giving a higher density (see also Behar et al., 2018, upper panel in Fig. 7). A similar overshoot for the magnetic field has also been seen in simulations (upper left panel of Fig. 1 Lindkvist et al., 2018). The gyroradii of protons in the $200 - 400\,\mathrm{km\,s^{-1}}$ range are $100 - 200\,\mathrm{km}$ in a $20\,\mathrm{nT}$ magnetic field. This is comparable to the thickness of the infant bow shock. The typical length scale of the structure in the upper left corner of Fig. 5 is about $10^3\,\mathrm{km}$, corresponding to approximately 2 gyroradii in the weaker magnetic field ($\sim 10\,\mathrm{nT}$) in that region.

This structure is seen to extend farther into the $-E_c$ than the $+E_c$ hemisphere. However, it is also seen in Fig. 5 that the box showing the area covered by Rosetta barely reaches the IBS in that hemisphere. Instead we would actually expect to observe events in the $+E_c$ hemisphere. This is what is also seen in the observations shown in the right panel of Fig. 4. We have said above that the plasma and boundaries in it, such as the IBS, are necessarily non-stationary for Rosetta to be able to observe them. Nevertheless, a picture emerges of a plasma that is structured much like the stationary images obtained from the simulation results, but with deformation and translation of the structure being driven by changes in the solar wind.

Despite searching the entire dataset for which proton data is available, most events are found at intermediate gas production rates (see Fig. 3), but where we do observe higher gas production rates, the warm protons are observed further from the nucleus than for low gas production rates. The IBS is a structure that was speculated to form only at intermediate gas production rates, where the mass-loading is sufficient to slow down the solar wind, but not significant enough to form a large bow shock. Thus the IBS location does agree with our findings for warm protons.

In order to provide proof that a boundary in a plasma is a shock, usually Rankine-Hugoniot are evaluated. However, the plasma environment of the comet is far from a single fluid MHD plasma where the R-H conditions could be used to investigate the transition. Such an approach has been employed in the past in the analysis of the Giotto flybys of comets 1P/Halley and 26P/Grigg–Skjellerup (Coates et al., 1990, 1997). Kessel et al. (1994) expanded the fluid theory to include effects of multiple ion species. For our situation, multi-ion and kinetic scale effects, and the non-stationarity of the shock need be accounted for. Motschmann et al. (1991a, b) derive multi-fluid R-H conditions and investigate the consequences of the second ion population on the behaviour of both ion flows at the shock. Fahr and Siewert (2015) show that including the kinetic effects from a multi-ion plasma changes the R-H conditions. They also found that the two ion populations can have a different behaviour when crossing

the shock, for example the protons are still supersonic downstream of the shock, while the electrons and pick up ions are not. This demonstrates the complexity of multi-ion shocks. This seems to be the case here as well. While the solar wind ions are shocked, the plasma density used here is derived from electron densities which are a mixture of solar wind electrons and photoelectrons. Thus, the overall plasma density does not change at the shock, which means that for the calculation of the R-H

conditions one should actually use an estimate of the solar wind ion density instead. Unfortunately this is not available. The proton density estimate from ICA shows a large scatter, of which some may be an instrumental effect thus we cannot use the data to test the shock conditions. Most models of shocks assume that the shock is in a stationary state. As already mentioned above, this is not the case for the observations with Rosetta, because to observe the transition the shock needs to move over the spacecraft and thus is not in a stationary state.

Omidi and Winske (1987) conducted one-dimensional hybrid simulations with the aim of modelling the spacecraft encounters with comets 1P/Halley and 21P/Giacobini-Zinner. They found that for oblique interaction (cone angle $55°$), shocklets form in a region of large amplitude wave activity. These shocklets convect downstream, where they break up due to dispersion, and new ones form further upstream. Thus, the process is repeated in a way that resembles shock reformation at planets (e.g. Balogh and Treumann, 2013). Although it is possible that shocklets form and shock reformation occurs also at comet 67P under certain

conditions, it is not the cause of the observations reported here. The shock encounters shown in Figs. 1, 6, A1, and A2 do not display the repetitive transitions in a wave-dominated region that would be expected for the shocklets reported by Omidi and Winske (1987).

Behar et al. (2017) reported similar features in the ICA data as those used for our event detection. They made an attempt to describe the observations with a simple ion test particle picture. The model produces results similar to those of the hybrid

simulations and fits with the deflection of the solar wind ions. It predicts a "caustic", an intersection of particle trajectories, as the boundary between the solar wind ion cavity and the solar wind dominated plasma. Our study concerns a similar structure, but we investigate here for the first time, the response of the plasma to such a change in the proton energy and thus broadens our understanding of the plasma. Although the test particle description agrees well with the deflection of the ions, it does not explain the heating of the ions and electrons. This is expected for a model that does not include the feedback of the particle

motion on the fields. However, it is entirely possible that those events that do not exhibit heating of the electrons are more similar to a model that treats the ions only as test particles with little influence on the behaviour of the plasma, while still including both electric and magnetic fields.

In future research, simultaneous observations at multiple points in space (Götz et al., 2019) would be a major advance in the ability to distinguish spatial and temporal dependence, and it is a necessity in assessing the stationarity or non-stationarity of the

375 infant bow shock. ESA's new F-class mission, Comet Interceptor, will be the first mission to provide multi-point measurements needed for this (Snodgrass and Jones, 2019), depending on conditions at the target comet. In hybrid simulations, the IBS response to solar wind variability could both shed light on the physics of cometary–solar wind interaction and aid in the interpretation of spacecraft data. There is, however, a vast range of cometary and solar wind parameters that must be sampled to obtain a complete picture. To understand the microphysics of shocks in general and infant bow shocks in particular one

must also progress from hybrid simulations to simulations that accurately model electrons as well as ions. To address the

microphysics, using spacecraft-based instruments, it is likewise important to resolve electron scales, both temporal and spatial. This could be done in 3D fully kinetic simulations.

## 5 Conclusions

We have expanded the previous study of the infant bow shock (Gunell et al., 2018) by searching for intervals with warm protons in the plasma around 67P. We first examined 13 cases in detail, and we have performed a statistical study of the whole Rosetta dataset. The results from both are similar. On average, the electron flux is increased in the downstream region, while the magnetic field magnitude decreases, and the magnetic field power spectral density and the density increase. All parameters, except for the magnetic field magnitude, behave according to what was defined for the infant bow shock. About 10% of all events identified here behave exactly as expected for the IBS. Since the Rosetta spacecraft moved at speeds of only approximately $1\,\mathrm{m\,s^{-1}}$, it is only when the infant bow shock reacts to changes in the upstream plasma that the spacecraft crosses it. Our large statistical dataset therefore includes observations not only of IBS crossings but also detections of shocked plasma downstream of the IBS itself, which nevertheless confirms its presence upstream of the spacecraft position. We therefore conclude that these warm protons are associated with the IBS in a very non-stationary plasma.

Most detections took place between $Q = 10^{26}\,\mathrm{s^{-1}}$ and $Q = 6 \times 10^{27}\,\mathrm{s^{-1}}$, which approximately corresponds to heliocentric distances between $1.7\,\mathrm{AU}$ and $2.7\,\mathrm{AU}$. It was observed preferentially in the $+E_c$ hemisphere due to its asymmetry.

While the precise nature of the IBS and the physics causing its formation remain to be revealed in future studies, we conclude that it is an asymmetric structure with many shock-like traits that is observed persistently during intermediate outgassing conditions. It may be that the infant bow shock is the low gas production rate manifestation of what becomes the more developed cometary bow shock as observed at larger comets such as Halley. This observation demonstrates again the uniqueness of the laboratory that the cometary environment can provide for the larger scope of plasma physics.

*Data availability.* Datasets of the RPC and ROSINA instruments onboard Rosetta are available at the ESA Planetary Science Archive (http://archives.esac.esa.int/psa Besse et al., 2018).

## Appendix A: Additional events

*Author contributions.* CG performed the analysis and selection of intervals in collaboration with HG. HN contributed the interpretation of the ICA observations, KL that of the IES observations, FJ that of the LAP observations. KHG and MGGTT contributed on the interpretation of the results. All authors contributed to the writing of the final manuscript as well as the interpretation of the data.

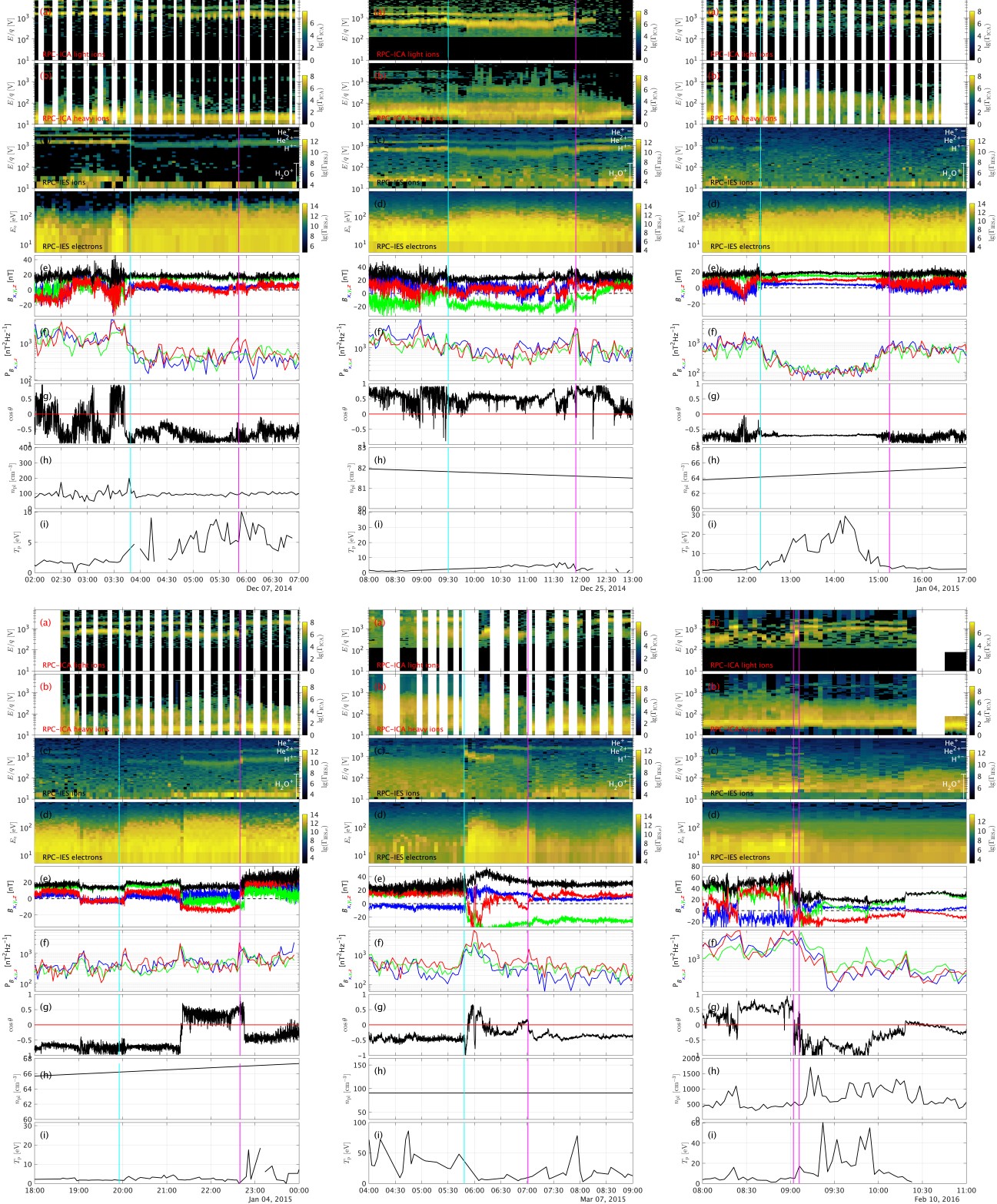

**Figure A1.** Observations of the plasma for the events shown in Table 1. Format is the same as in Figure 1.

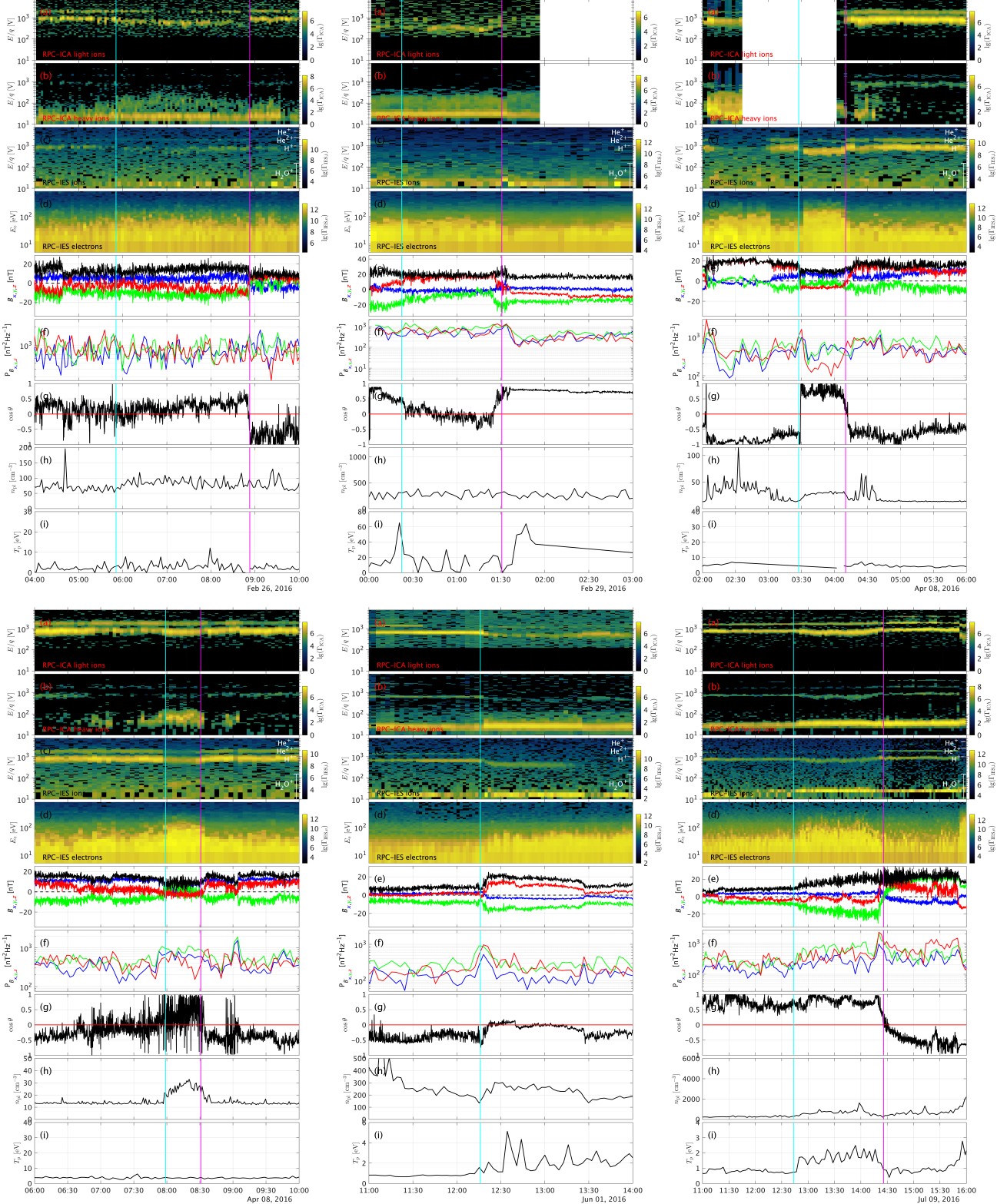

**Figure A2.** Observations of the plasma for the events shown in Table 1. Format is the same as in Figure 1.

*Competing interests.* None of the authors have any competing interests.

*Acknowledgements.* Part of this work was supported by the DLR project number 50 QP 1401. CG is supported by an ESA Research Fellowship. HG was supported by the Swedish National Space Agency grant 108/18 and by the Belgian Science Policy Office through the Solar-Terrestrial Centre of Excellence. Datasets of the RPC and ROSINA instruments onboard Rosetta are available at the ESA Planetary Science Archive (http://archives.esac.esa.int/psa Besse et al., 2018).

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
