# Peer review of "Warm protons at comet 67P/Churyumov-Gerasimenko – Implications for the infant bow shock"

_Annales Geophysicae, 2020_

## Referee Comment (RC1) · Anonymous Referee #1 · 6 Nov 2020

This study performed a survey of warm protons at 67P which are studied due to their association with a newly formed bow shock. I think this survey has value. Such large data surveys are highly informative for qualitatively and quantitatively characterizing new environments and it is appreciated that these can be difficult to draw conclusions from when using a single spacecraft with no upstream monitor. I, however, do not believe the hypothesis or focus of this investigation is particularly well defined and, as it is currently written, the conclusions of the study are therefore not supported by the data presented. I am therefore unwilling to recommend publication until the following points are addressed:

- Line 65: More justification is needed as to why the warm proton flux is used as the main signal of the shock over other parameters, or over being used in conjunction with

other parameters. The discussion mentions reasons as to why other parameters are not sufficient, i.e. hot electrons might be missed. Such arguments should be moved up in the manuscript and used as an assumption and justification for using the warm protons. What other phenomena could produce warm protons?

- Line 155: The selection criteria is outlined as a 'pronounced decrease' in the proton energy and a 'broadening of the energy band'. Figure 2a however shows values above and below unity, does this not contradict the selection criteria? If $v_{m,h}$ in Figure 2 is different to the H+ (E/q) column in Table 1 this needs to be described. Some clarity is needed here.

- Line 200: Statistics such as Figure 2 (can (a) be quantitatively represented on line 200 with a percentage?), 60% electron energy increase on line 205, 68% on line 210 for the B field decrease, 52% on line 210 for the s/c potential decrease, indicate to me that the detection criteria for the Infant Bow Shock is not sufficient – i.e. the shock has not been persistently detected.

- Line 40: There isn't mention of an induced magnetopause or the plasma boundaries referred to in the first line of the abstract. Surely this should follow on from the introduction to how an observable shock forms due to increased mass loading. Where is the induced magnetopause expected to form in relation to where these events are detected? Could these events be associated with this instead?

- Line 160: The use of the spacecraft potential is I believe not appropriate. That the spacecraft potential follows the density is only true when assuming a constant temperature. This study is focused on studying a shock which will alter and heat the plasma.

- Line 240: This sentence is an evaluation of the hypothesis of whether these events are indeed the shock. 250 to 270 then appears to go into some, perhaps reasonable but certainly speculative, arguments about why the statistics are not good. The study therefore, with reference to line 345 and 355, seems to go on to imply that all warm proton detections are associated with the shock. This may be true but is not supported

by the analysis presented. See point below:

- Line 345: The sentence "All parameters, except for the magnetic field magnitude, behave according to what was defined for the infant bow shock" doesn't seem to me to agree with the statistics presented in Figure 2 and the next sentence which states that only 10% agree. Some clarity is needed here.

- Line 355: The study concludes that the shock is an asymmetric structure persistently observed. This to me is in contradiction to the statistical results - surely the conclusions are that the shock is not persistently detected given many of the statistics are not much better than tossing a coin. It is this aspect of the paper that I feel is biased in a certain direct and therefore draws conclusions not fully supported by the data.

- Line 245 states that some events could be examples of phenomena other than a weak shock. What are the comparative phenomena? Line 130 mentions some other effects such as changes in solar wind velocity/density/cometary ion density but these aren't adequately described or referenced. This discussion on alternate production mechanisms for warm protons is central to the arguments being made in this manuscript and ties back to my first point regarding Line 65.

In addition to the above conceptual points I noticed the below points as I read through the manuscript which also should be addressed:

- Line 220: Heliocentric distance I think should be mentioned before now to orient the reader. How are the 370 events selected? How many Rosetta orbits are there in total outside the cavity? How many with good data that the 370 are selected from? Some explanation and context is needed here for reproducibility. I recommend changing Line 110 to include this information.

- Line 40: Can the "few percent" be better constrained or referenced with respect to 67P?

- Line 70: Should "easily" be "earlier" instead?

- Line 100: Can you please reference the "is believed".

- Line 130: What publications does the "(as stated in previous publications)" refer to?

- Line 159: Why is the following sentence interesting? "Interestingly, the flux diminishes at the same time that the proton energy increases gradually".

- Line 265: The description of the solar wind "pushing" the shock is not accurate, shocks are modes as opposed to pressure balanced structures. Rewording is needed.

- Line 308: I do not believe the authors have "made attempts to conclusively show that this structure is indeed a shock in the fluid dynamics sense." This is far too strong and I think would only be correct if sufficient effort had been made to deal with the RH jump conditions.

---

## Referee Comment (RC2) · Anonymous Referee #2 · 7 Dec 2020

This is an interesting and detailed paper analysing over 300 more examples of the 'infant bow shock', a feature observed in Rosetta RPC data, following initial analysis of 2 events by Gunnell et al., 2018. The analysis principally considers that warmer, slower protons seen in the 'downstream' region of some of the examples are the main characteristics of the feature, although higher electron flux, lower magnetic field, higher oscillations in the magnetic field, and higher density (usually) were also characteristics of the region with warmer protons. The authors conclude that the plasma characteristics in the warmer proton regions are associated with 'intermediate' production rates in the Rosetta data, and are usually seen in the positive convention electric field hemisphere. They suggest that the 'infant bow shock' is an asymmetric structure and may develop into an 'ordinary' bow shock observed at stronger comets. The paper is well

written and the data analysis careful, but the current version suffers from a lack of references on earlier work on missions prior to Rosetta and some confusing features; it would be suitable for publication after some revision.

Major points for additional analysis and comment 1. The principal diagnostic is the observation of 'warmer, slower' protons, but this is not quantified in the paper as much as it could be, although visible in spectrograms. Some simple 1D analysis (building on the $v_{m,H}$ shown here) would allow calculation of the velocity, but the main suggestion here is that at least some analysis and characterisation of the width of the proton spectra, and the jump across the feature, would provide a quantitative indication related to temperature, which is missing from the current analysis although it is a prime diagnostic. 2. Some calculations of Mach number based on the analysis of Smith et al (1986) for comet GZ and Coates et al (1990, 1997) could be attempted for at least some of the observed 'infant bow shock' features in the data, as well as in the related simulations. This would strengthen the use of the word 'shock', and allow comparison to 'shocklets' seen in other simulations (e.g. Omidi et al). The change in velocity, magnetic field and density could be estimated sufficiently to do this. 3. In Figure 3, some of the $v_{m,H}$ values indicate an increase of velocity from upstream to downstream – this seems counter-intuitive for any shock

Minor points Line 23 – the text refers to a 'fully formed shock' at comets, but has this been observed by Rosetta? The references provided all relate to Rosetta. Additional references include Smith et al, 1986, Coates et al, 1990, 1996, relating to GZ, Halley and GS. Line 28 – Mass loading, deceleration and deflection were all aspects of earlier studies on Giotto and AMPTE data which are not referenced here (Coates et al., 2015, and references therein, are relevant) Line 38 – the convective electric field upstream of the comet drives the pickup process as shown in earlier studies (e.g. Neugebauer et al., 1989, Coates et al., 1990 and many other studies Line 45 – the bow shock location, formation and features have been studied in detail using data from Giotto by others also (e.g. Coates et al., 1990, 1996) Lines 54-55 – Bow shock studies at comets

and other solar system objects have been more extensive than the references would indicate Line 64 – 'proton velocity distribution becomes broader and the bulk velocity decreases' – visible in the spectrograms usually, but needs some quantification (see comment 1 above) Line 66 typo 'ensure' Line 69 – please specify the 'similarity to a bow shock at a fully developed comet', using references from earlier missions – which changes were seen before and which are different here Line 83 – 'Often, the signal is still visible in the RPC-IES instrument' – presumably due to different FOV, please add a comment Line 93 – 'partially complementary to ICA' – please specify the fields of view and extent of overlap/complementarity Line 115 – 'need to be at significantly lower energies' – please quantify Line 134 – 'as stated in previous publications' – references and precision needed Table 1 -would be useful to define and include a parameter/measurement associated with the width of the proton distribution, especially as this is one of the major diagnostics of the events (again, see comment 1 above) Line 157 – It is interesting that the alpha particle and He+ spectra follow the proton distributions yet both remain distinct, another indication that the transitions are weak, a comment could be added on this Line 163 – More precise to say 'is more negative' rather than 'lower' Line 164-5 – 'the lower the spacecraft potential, the higher the density' could be reworded 'higher plasma density would increase the flux of electrons to the spacecraft, providing more negative spacecraft potentials' Line 165 – 'This the density is higher' – how much higher, and where? How is this visible in the data shown? Fig 2 caption – add comment (see definitions in text), or add a short explanation for the definition of the parameters shown Line 178 – 'transition can sometimes be very broad' – can this be quantified e.g. with respect to the electron, proton and heavy ion gyroradius? (see e.g. Coates et al. 1990) Line 190 typo 'where' Line 200-215 – the authors could usefully define and calculate a parameter associated with the width of the proton distributions (as with the velocity change vm,H this is a key indicator) – see also comment 1 above Section 3.3 general comment – is there any evidence for larger/more developed jumps with increasing Q? Line 224 – as well as Deca et al, there were earlier papers on momentum balance in the AMPTE releases and in comets (see Coates et al. 2015,

and references therein, eg Coates et al, JGR 1986, Johnstone et al., Geophys Monograph 38, 1985, Coates et al, Adv Space Res 1988)) Line 238 – 'protons with higher temperatures' - this should be quantified, see comment 1 Line 242 – 'flux of electrons does increase downstream' – might some of this be associated with spacecraft potential changes? Line 250 - 'different for electrons and protons' – and heavy ions? Line 254 – Might shocklets (e.g. Omidi et al.), and/or upstream cavities, be relevant Line 262 – 10s of minutes – how might this compare to gyroperiods/radii? Line 274 – please specify/clarify/indicate on Fig 6 the times discussed (first/second half) Line 283 – 'density of the plasma does not change significantly' – if anything, the spacecraft potential is more negative, thus density higher, in the 'upstream' region in this case Line 285 – could calculate the ratio between the solar wind and the local plasma density Line 288 – it would be useful to mention the assumed gas production rate Q in simulation and for the relevant observation Line 290 – please indicate the suggested 'IBS' location on Fig 5 Line 294 – what is the scale of proton gyration compared to the features seen in the simulation Line 299 – Does $+E_c$ correspond to $E_{parallel z}$ as on the Figure? Line 306 – 'not significant enough to form a large bow shock' – rather than 'large' do you mean fully developed? Might there be a relation to shocklets? Line 310 – Kessel et al (JGR, 1994) also reformulated the jump conditions and determined shock normal for multiple ion shocks Line 322 – Re shock motion – as mentioned above, it should be possible to estimate the shock motion speed from the change in velocity and shock normal (e.g. Smith et al, Coates et al) Line 325 – please briefly explain the term 'caustic' Line 334 – re Comet Interceptor, depending on the gas production rate of the target comet, any observed cometary bow shock may be more fully developed than the features discussed here Line 340 – also, 3D fully kinetic simulations would be valuable Line 345 – refers to a 'density proxy' – is this the spacecraft potential? In Fig 6 the density appears higher upstream Line 355 – More accurate to say 'It may be that the 'infant bow shock' is the low production rate manifestation of what becomes the more developed cometary bow shock as observed at larger comets such as Halley' (add references). Also discuss shocklets in this context Line 357 – 'ordinary' may not be the correct adjective for the

complex bow shock structure, with changes at proton and heavy ion gyroscales, as observed at comets such as Halley (e.g. Coates et al., 1987)

---

## Referee Comment (RC3) · Anonymous Referee #3 · 22 Dec 2020

This paper has analyze the data from multiple instruments observations upstream and downstream of the infant bow at the comet CG when the production rate is intermediate. The results are interesting and in my view is worth publishing after some minor changes. Comments and suggestions are listed below.

Major comments: 1. Near Line 70, in this paper you are mainly exploring the characteristics in the data when the spacecraft crossed the infant shock. Can you also briefly mention and cite some references on what the data will be like if an ordinary or classical shock is crossed, so that readers can easily see the similarities and differences between the infant shock and the ordinary shock.

Minor comments: 1. Line 2: after "infant bow shock" add "(IBS)".

[Figure]

2. Line 25: "and with it the amount of ice" -> "with increasing amount of ice"

3. Line 40: lower gyroradii -> smaller gyroradii

4. Line 49: "the comet's, frame of reference" -> "the comet's frame of reference"

5. Line 66: "insure" -> "ensure"

6. Line 74: "it's characteristics" -> "its characteristics"

7. Line 119: "instead" -> "because" ?

8. Line 138: by-eye inspection -> inspection by eyeball ?

9. Line 159 & 160: "Interestingly, the flux diminishes at the same time that the proton energy increases gradually." Can you add some theoretical explanation to this phenomena?

10. Line 163: the angle between the x-axis and magnetic field -> the angle between the x-axis and electric field?

10. Line 166: Does spacecraft attitude mean spacecraft orientation? Can you explain what are alpha_{x,y,z} of the spacecraft attitude?

11. Line 173: "we find that the energy of the electrons is almost always increased". Is it consistent with your expectations? Can you add explanation the increase of electron energy and decrease in ion energy?

12 Line 201: below than above unity -> below unity?

13 Line 244: The statement "at least some of this discrepancy might be attributable to the inability of the flux at 60eV or 120eV to accurately represent the electron spectra" is not clear to me. Can you elaborate this point?

---

## Author Comment (AC1) · 26 Feb 2021

**Response to Referee #1**
**Warm protons at comet 67P/Churyumov-Gerasimenko – Implications for the infant bow shock**

We thank the referee for the comments and suggestions. We have made the necessary amendments to the paper and answers to comments may be found below. (Blue: Referee comment, black: our answer).

Line 65: More justification is needed as to why the warm proton flux is used as the main signal of the shock over other parameters, or over being used in conjunction with other parameters. The discussion mentions reasons as to why other parameters are not sufficient, i.e. hot electrons might be missed. Such arguments should be moved up in the manuscript and used as an assumption and justification for using the warm protons. What other phenomena could produce warm protons?
Line 245 states that some events could be examples of phenomena other than a weak shock. What are the comparative phenomena? Line 130 mentions some other effects such as changes in solar wind velocity/density/cometary ion density but these aren't adequately described or referenced. This discussion on alternate production mechanisms for warm protons is central to the arguments being made in this manuscript and ties back to my first point regarding Line 65.
We added the following after the sentence on lines 65-67 (line numbers referring to the previous version of the manuscript): "According to Balogh & Treumann (2013) , the slowing down and heating of the medium over a narrow layer or boundary is the defining feature of any shock." Near line 130 (again old numbering) we added a few words of explanation by changing "can also cause warm protons to appear" to "can also move the boundary, causing warm protons to appear at the spacecraft". We also added a reference.

Line 155: The selection criteria is outlined as a 'pronounced decrease' in the proton energy and a 'broadening of the energy band'. Figure 2a however shows values above and below unity, does this not contradict the selection criteria? If vm,h in Figure 2 is different to the H+ (E/q) column in Table 1 this needs to be described. Some clarity is needed here.
The solar wind energy spectra measured by Rosetta are difficult to condense into a reliable parameter that can be used for statistics. Further investigation has revealed that often the $v_{m,H}$ parameter is not a good representation of what is visible in the spectra, due to cross-talk, field of view and noise issues. In this updated version of the paper we therefore have omitted the statistical study of the $v_{m,H}$ parameter, as the results are not reliable.

Line 200: Statistics such as Figure 2 (can (a) be quantitatively represented on line 200 with a percentage?), 60% electron energy increase on line 205, 68% on line 210 for the B field decrease, 52% on line 210 for the s/c potential decrease, indicate to me that the detection criteria for the Infant Bow Shock is not sufficient – i.e. the shock has not been persistently detected.
For the statistics on the velocity, see answer above. As stated in Section 2.2, we do not use the magnetic field, or the s/c potential (now density) to identify an infant bow shock. We simply are looking at where these transitions from a cold, fast solar wind to a warm, slow solar wind occur. The statistics presented here are just a result, and not a criterion.

Line 40: There isn't mention of an induced magnetopause or the plasma boundaries referred to in the first line of the abstract. Surely this should follow on from the introduction to how an observable shock forms due to increased mass loading. Where is the induced magnetopause expected to form in relation to where these events are detected? Could these events be associated with this instead?
In our current understanding of the comet's interaction with the solar wind, no induced magnetopause exists. This concept is often used at Mars/Venus, but it is not used at the comet. The boundaries that are observed at the comet are a solar wind ion cavity (e.g. Nilsson et al. 2017 MNRAS), a diamagnetic cavity (e.g. Goetz et al 2016 A&A) and there are some indications of

a collisionopause, i.e. a tenuous boundary where collisional coupling of ions or electrons to the neutral gas become important (Mandt et al 2016 MNRAS). The diamagnetic cavity and the collisionopause lie within the solar wind ion cavity and the solar wind ion cavity by definition is the region that does not contain any solar wind ions. Thus, the boundary that we observe here lies outside of all of these boundaries and is distinct from them. We have added a paragraph in the introduction similar to this explanation.

*Line 160: The use of the spacecraft potential is I believe not appropriate. That the spacecraft potential follows the density is only true when assuming a constant temperature. This study is focused on studying a shock which will alter and heat the plasma.*

Johansson et al (2020) showed that the Rosetta spacecraft potential is predominantly dependent on density and only found a weak dependence on temperature of the electrons. It is thus a good indicator of density even when the temperature changes. Since the writing of this manuscript a more reliable density dataset has become available and thus, for this version of the manuscript, we changed from spacecraft potential to density. As expected, this does not change the results. The implications of these statistics are then discussed in the next paragraphs.

*Line 240: This sentence is an evaluation of the hypothesis of whether these events are indeed the shock. 250 to 270 then appears to go into some, perhaps reasonable but certainly speculative, arguments about why the statistics are not good. The study therefore, with reference to line 345 and 355, seems to go on to imply that all warm proton detections are associated with the shock. This may be true but is not supported by the analysis presented. See point below: Line 345: The sentence "All parameters, except for the magnetic field magnitude, behave according to what was defined for the infant bow shock" doesn't seem to me to agree with the statistics presented in Figure 2 and the next sentence which states that only 10% agree. Some clarity is needed here.*

The apparent contradiction comes from the statement about the 10 %. We have revised that sentence to clarify: "In about 10 % of the cases all parameters that were evaluated, the magnetic field included, behave as expected at the same time."

*Line 355: The study concludes that the shock is an asymmetric structure persistently observed. This to me is in contradiction to the statistical results - surely the conclusions are that the shock is not persistently detected given many of the statistics are not much better than tossing a coin. It is this aspect of the paper that I feel is biased in a certain direction and therefore draws conclusions not fully supported by the data.*

As discussed in the text for $v_{m,H}$, this parameter is not very well suited for usage in a statistical study. It needs to be treated with caution, which we have done for the smaller subset of events, but chose not to do for the large dataset. A reexamination of the parameter has reaffirmed this. To avoid confusion, this parameter was removed from figure 3 and is only discussed in the text now. The defining features of a shock, which we have used to select the events, are the heating and slowing down of the streaming plasma. This applies for all the observed cases for the ion spectra. The other quantities, shown in Fig. 2, should be seen as descriptive rather than defining, and as such do not invalidate the conclusions. This was also made clearer already in the abstract.

*Line 220: Heliocentric distance I think should be mentioned before now to orient the reader. How are the 370 events selected? How many Rosetta orbits are there in total outside the cavity? How many with good data that the 370 are selected from? Some explanation and context is needed here for reproducibility. I recommend changing Line 110 to include this information.*

As Rosetta is near an object with very low gravity, most of the time there is no bound orbit. In principle, Rosetta is orbiting the Sun along side the comet. The trajectory is highly complex and cannot be quantified simply as orbits within a certain region. We have used the entire dataset, as stated in Section 2.2, to find these events. Figure 3 shows the cometocentric distance for the entire mission in grey, thus we do provide information on the spacecraft-comet separation.

Figure 4 (left panel) shows the coverage of Rosetta in the CSE frame. The PSA was referenced and a link to the spice kernels is provided in the referenced Rosetta PSA website. It is therefore possible to reproduce everything shown here, including the event locations, as given in the supplementary material.

Line 40: Can the "few percent" be better constrained or referenced with respect to 67P?

For the event from Figure 1 we can derive a proton density of ca $0.5\,\mathrm{cm}^{-3}$ from the ICA moments. The plasma density is of the order of $1000\,\mathrm{cm}^{-3}$, this would mean a proton fraction of 0.05%. This seems rather low, however, the proton density estimate is also extremely low. Even assuming that ICA underestimate by a factor of 10, would only give us a fraction of 0.5%. Thus, for the larger plasma dynamics, the protons can be neglected. Alternatively we can make an estimate of the maximum proton density based on a simple fluid model, which seems a better way to get the maximum fraction of protons in the plasma. We have added to the paper: " We can estimate the fraction of cometary ions for the event shown in Fig. 1. The cometary ion density is of the order of $1000\,\mathrm{cm}^{-3}$ and we can estimate the maximum proton density from a simple back-of-the-envelope calculation: assuming a solar wind density of $3\,\mathrm{cm}^{-3}$ (typical for heliocentric distances around 2 AU) and a compression factor of $\sim 4$, we get a proton density of $12\,\mathrm{cm}^{-3}$. This is close to what is also observed in the simulation used below. This gives a fraction of $\sim 99\%$ cometary ions. Even if this estimate is very rough, it is clear that the cometary ions are at this point clearly dominating the plasma and the solar wind has only very little influence on the plasma density."

Line 70: Should "easily" be "earlier" instead?

Indeed, this was corrected.

Line 100: Can you please reference the "is believed".

Yes, explanations can be found in Johansson et al 2020, which was added to the text.

Line 130: What publications does the "(as stated in previous publications)" refer to?

This refers to Gunell et al 2018 and is now explicitly mentioned in the text.

Line 159: Why is the following sentence interesting? "Interestingly, the flux diminishes at the same time that the proton energy increases gradually."

We have clarified the statement: "Interestingly, the flux diminishes at the same time that the proton energy increases gradually, implying that the spacecraft moved slowly upstream in a shock-fixed frame of reference into a region with less electron heating and a less slowed-down proton distribution."

Line 265: The description of the solar wind "pushing" the shock is not accurate, shocks are modes as opposed to pressure balanced structures. Rewording is needed.

This is indeed correct. We reworded the sentence: "One possibility is that an increase in the solar wind dynamic pressure increases the mass-loading threshold of the plasma (Biermann et al 1967) which means that the critical condition for a shock is met later in the flow, and thus closer to the comet. "

Line 308: I do not believe the authors have "made attempts to conclusively show that this structure is indeed a shock in the fluid dynamics sense." This is far too strong and I think would only be correct if sufficient effort had been made to deal with the RH jump conditions.

In a first attempt, we indeed tried to evaluate the R-H conditions for one event. However, as detailed in the text, this is a moot point because the R-H conditions are not applicable here. But, in order to avoid any misunderstandings, this sentence was changed to a more generic: "
[revised manuscript text omitted]

A. J. Coates, A. D. Johnstone, R. L. Kessel, D. E. Huddleston, and B. Wilken. Plasma parameters near the comet Halley bow shock. *J. Geophys. Res.*, 95:20701–20716, December 1990. doi: 10.1029/JA095iA12p20701.

A. J. Coates, A. D. Johnstone, and F. M. Neubauer. Cometary ion pressure anisotropies at comets Halley and Grigg-Skjellerup. *J. Geophys. Res.*, 101(A12):27573–27584, December 1996. doi: 10.1029/96JA02524.

A. J. Coates, C. Mazelle, and F. M. Neubauer. Bow shock analysis at comets Halley and Grigg-Skjellerup. *J. Geophys. Res.*, 102(A4):7105–7113, April 1997. doi: 10.1029/96JA04002.

C. Goetz, C. Koenders, K. C. Hansen, J. Burch, C. Carr, A. Eriksson, D. Frühauff, C. Güttler, P. Henri, H. Nilsson, I. Richter, M. Rubin, H. Sierks, B. Tsurutani, M. Volwerk, and K. H. Glassmeier. Structure and evolution of the diamagnetic cavity at comet 67P/Churyumov-Gerasimenko. *MNRAS*, 462:S459–S467, November 2016a. doi: 10.1093/mnras/stw3148.

C. Goetz, C. Koenders, I. Richter, K. Altwegg, J. Burch, C. Carr, E. Cupido, A. Eriksson, C. Güttler, P. Henri, P. Mokashi, Z. Nemeth, H. Nilsson, M. Rubin, H. Sierks, B. Tsurutani, C. Vallat, M. Volwerk, and K.-H. Glassmeier. First detection of a diamagnetic cavity at comet 67P/Churyumov-Gerasimenko. *A&A*, 588:A24, April 2016b. doi: 10.1051/0004-6361/201527728.

Charlotte Götz, Herber Gunell, Martin Volwerk, Arnaud Beth, Anders Eriksson, Marina Galand, Pierre Henri, Hans Nilsson, Cyril Simon Wedlund, Markku Alho, Laila Andersson, Nicolas Andre, Johan De Keyser, Jan Deca, Yasong Ge, Karl-Heinz Glaßmeier, Rajkumar Hajra, Tomas Karlsson, Satoshi Kasahara, Ivana Kolmasova, Kristie LLera, Hadi Madanian, Ingrid Mann, Christian Mazelle, Elias Odelstad, Ferdinand Plaschke, Martin Rubin, Beatriz Sanchez-Cano, Colin Snodgrass, and Erik Vigren. Cometary Plasma Science – A White Paper in response to the Voyage 2050 Call by the European Space Agency. *arXiv e-prints*, art. arXiv:1908.00377, Aug 2019.

Herbert Gunell, Charlotte Goetz, Cyril Simon Wedlund, Jesper Lindkvist, Maria Hamrin, Hans Nilsson, Kristie Llera, Anders Eriksson, and Mats Holmström. The infant bow shock: a new frontier at a weak activity comet. *A&A*, 619:L2, November 2018. doi: 10.1051/0004-6361/201834225.

F. L. Johansson, A. I. Eriksson, N. Gilet, P. Henri, G. Wattieaux, M. G. G. T. Taylor, C. Imhof, and F. Cipriani. A charging model for the Rosetta spacecraft. *A&A*, 642:A43, October 2020. doi: 10.1051/0004-6361/202038592.

R. L. Kessel, A. J. Coates, U. Motschmann, and F. M. Neubauer. Shock normal determination for multiple-ion shocks. *J. Geophys. Res.*, 99(A10):19359–19374, October 1994. doi: 10.1029/94JA01234.

C. Koenders, K.-H. Glassmeier, I. Richter, U. Motschmann, and M. Rubin. Revisiting cometary bow shock positions. *Planetary and Space Science*, 87:85–95, October 2013. doi: 10.1016/j.pss.2013.08.009.

K. E. Mandt, A. Eriksson, N. J. T. Edberg, C. Koenders, T. Broiles, S. A. Fuselier, P. Henri, Z. Nemeth, M. Alho, N. Biver, A. Beth, J. Burch, C. Carr, K. Chae, A. J. Coates, E. Cupido, M. Galand, K.-H. Glassmeier, C. Goetz, R. Goldstein, K. C. Hansen, J. Haiducek, E. Kallio, J.-P. Lebreton, A. Luspay-Kuti, P. Mokashi, H. Nilsson, A. Opitz, I. Richter, M. Samara, K. Szego, C.-Y. Tzou, M. Volwerk, C. Simon Wedlund, and G. Stenberg Wieser. RPC observation of the development and evolution of plasma interaction boundaries at 67P/Churyumov-Gerasimenko. *MNRAS*, 462:S9–S22, November 2016. doi: 10.1093/mnras/stw1736.

F. M. Neubauer, K. H. Glassmeier, M. Pohl, J. Raeder, M. H. Acuña, L. F. Burlaga, N. F. Ness, G. Musmann, F. Mariani, M. K. Wallis, E. Ungstrup, and H. U. Schmidt. First results from the Giotto magnetometer experiment at comet Halley. *Nature*, 321:352–355, May 1986. doi: 10.1038/321352a0.

H. Nilsson, G. Stenberg Wieser, E. Behar, H. Gunell, M. Galand, C. Simon Wedlund, M. Alho, C. Goetz, M. Yamauchi, P. Henri, and E. Odelstad A.I. Eriksson. Evolution of the ion environment of comet 67P during the rosetta mission as seen by RPC-ICA. *Monthly Notices of the Royal Astronomical Society*, 469 (Suppl_2):S252–S261, 2017. doi: 10.1093/mnras/stx1491.

N. Omidi and D. Winske. A kinetic study of solar wind mass loading and cometary bow shocks. *J. Geophys. Res.*, 92(A12):13409–13426, December 1987. doi: 10.1029/JA092iA12p13409.

C. Simon Wedlund, E. Behar, H. Nilsson, M. Alho, E. Kallio, H. Gunell, D. Bodewits, K. Heritier, M. Galand, A. Beth, M. Rubin, K. Altwegg, M. Volwerk, G. Gronoff, and R. Hoekstra. Solar wind charge exchange in cometary atmospheres III. Results from the Rosetta mission to comet 67P/Churyumov-Gerasimenko. *Astronomy & Astrophysics*, 2019. doi: 10.1051/0004-6361/201834881.

Edward J. Smith, Bruce T. Tsurutani, James A. Slavin, Douglas E. Jones, George L. Siscoe, and D. Asoka Mendis. International cometary explorer encounter with Giacobini-Zinner: Magnetic field observations. *Science*, 232(4748):382–385, 1986. ISSN 0036-8075. doi: 10.1126/science.232.4748.382.

**A    Additional events**

[Figure]

Figure 6: Observations of the plasma for the events shown in Table 1. Format is the same as in Figure 1.

[Figure]

Figure 7: Observations of the plasma for the events shown in Table 1. Format is the same as in Figure 1.

---

## Author Comment (AC2) · 26 Feb 2021

**Response to Referee #2**

**Warm protons at comet 67P/Churyumov-Gerasimenko – Implications for the infant bow shock**

We thank the referee for the constructive comments and suggestions. We have made the necessary amendments to the paper and answers to comments may be found below. (Blue: Referee comment, black: our answer).

The principal diagnostic is the observation of 'warmer, slower' protons, but this is not quantified in the paper as much as it could be, although visible in spectrograms. Some simple 1D analysis (building on the vm,H shown here) would allow calculation of the velocity, but the main suggestion here is that at least some analysis and characterisation of the width of the proton spectra, and the jump across the feature, would provide a quantitative indication related to temperature, which is missing from the current analysis although it is a prime diagnostic.

Line 64 – 'proton velocity distribution becomes broader and the bulk velocity decreases' – visible in the spectrograms usually, but needs some quantification (see comment 1 above)

Line 200-215 – the authors could usefully define and calculate a parameter associated with the width of the proton distributions (as with the velocity change vm,H this is a key indicator) – see also comment 1 above Section 3.3 general comment – is there any evidence for larger/more developed jumps with increasing Q?

Line 238 – 'protons with higher temperatures' - this should be quantified, see comment 1

Indeed, it would be good to look at the width/temperature of the protons as well as the spectra. We have used a temperature dataset, derived from the proton spectra, when initially selecting intervals. However, it turned out to be not very useful, because the correctness of the temperature depends on there being no cross-talk between mass channels in the ICA instrument as well as a good signal to noise ratio. Crosstalk is a well known issue and not easy to correct for, especially in an automatically generated data set. The attached figure shows the events from Figure 1 and 6 in the paper, with the temperature added. While the temperature describes the proton signature in the spectra very well for the event in July 2016, is does not do so for February 2016. In the ICA solar wind spectra, it becomes clear that there is significant cross-talk where signatures from cometary ions appear in the solar wind spectra. Thus we discarded this parameter for further statistical analysis. However, since the temperature dataset is suitable for a more thorough investigation of a smaller set of events, we have included the temperature in all figures and now discuss it in Sect. 3.1.

Some calculations of Mach number based on the analysis of Smith et al (1986) for comet GZ and Coates et al (1990, 1997) could be attempted for at least some of the observed 'infant bow shock' features in the data, as well as in the related simulations. This would strengthen the use of the word 'shock', and allow comparison to 'shocklets' seen in other simulations (e.g. Omidi et al). The change in velocity, magnetic field and density could be estimated sufficiently to do this.

Line 310 – Kessel et al (JGR, 1994) also reformulated the jump conditions and determined shock normal for multiple ion shocks

In the discussion, we touch very briefly on this point, but we see that this could be further elaborated. The assumptions made by Coates et al (1990) for the discussion of the shock are: low cometary ion density fraction (1.5%), single fluid 1D model with a mass source term in the continuity equation, a steady state. None of these apply for the case of 67P. At 67P at this stage the cometary ions dominate density wise and are as important as the solar wind momentum wise. They constitute not only a mass source, but also a momentum source (Nilsson et al 2020). The 1D approximation does not hold, because significant deflection of the solar wind and cometary ions has already happened by the time the solar wind reaches the region where Rosetta is located (see e.g. Behar et al 2016). As Rosetta can only observe a boundary when the boundary itself is moving, everything we observe is inherently not in a steady state.

Thus, the model used by Coates et al (1990) is not applicable here. The solution presented by Kessel et al (1994) does include the second ion population, but otherwise suffers from the same problems. Instead we looked at two-ion shocks that treat ions as particles and not fluids. The most applicable example is that of Fahr & Siewert (2015). As detailed in the text, we cannot get a good estimate of the proton density alone and thus it is not possible to actually test the model by Fahr & Siewert for this case.

In Figure 3, some of the vm,H values indicate an increase of velocity from upstream to downstream – this seems counter-intuitive for any shock

This is true. As discussed for the temperature above, and in the text for $v_{m,H}$, this parameter is not very well suited for usage in a statistical study. It needs to be treated with caution, which we have done for the smaller subset of events, but chose not to do for the large dataset. A reexamination of the parameter has reaffirmed this. To avoid confusion, this parameter was removed from figure 3 and is only discussed in the text now.

Line 23 – the text refers to a 'fully formed shock' at comets, but has this been observed by Rosetta? The references provided all relate to Rosetta. Additional references include Smith et al, 1986, Coates et al, 1990, 1996, relating to GZ, Halley and GS.Line 28 – Mass loading, deceleration and deflection were all aspects of earlier studies on Giotto and AMPTE data which are not referenced here (Coates et al., 2015, and references therein, are relevant)Line 38 – the convective electric field upstream of the comet drives the pickup process as shown in earlier studies (e.g. Neugebauer et al., 1989, Coates et al., 1990 and many other studiesLine 45 – the bow shock location, formation and features have been studied in detail using data from Giotto by others also (e.g. Coates et al., 1990, 1996)Lines 54-55 – Bow shock studies at comets and other solar system objects have been more extensive than the references would indicateLine 69 – please specify the 'similarity to a bow shock at a fully developed comet', using references from earlier missions – which changes were seen before and which are different here

We focused here more on the observations at 67P, but indeed a more extensive summary also of observations at other comets is beneficial. This was added.

Line 66 typo 'ensure'

Corrected

Line 83 – 'Often, the signal is still visible in the RPC-IES instrument' – presumably due to different FOV, please add a comment .

Yes, this is due to FoV effects. A comment was added.

Line 93 – 'partially complementary to ICA' – please specify the fields of view and extent of overlap/complementarity

The field of view is partially complementary. The exact FoV can be found in the ICA User Guide, available on the PSA. We have added a reference to this in the text.

Line 115 – 'need to be at significantly lower energies' – please quantify

The peak of the proton energy distribution needs to decrease by at least 60eV (corresponding to three ICA energy bins, at the lowest proton energies (250 eV)). This was added in the text.

Line 157 – It is interesting that the alpha particle and He+ spectra follow the proton distributions yet both remain distinct, another indication that the transitions are weak, a comment could be added on this

This is a good point, it was added.

Line 163 – More precise to say 'is more negative' rather than 'lower'

Agreed.

Line 164-5 – 'the lower the spacecraft potential, the higher the density' could be reworded 'higher plasma density would increase the flux of electrons to the spacecraft, providing more negative spacecraft potentials'

Since the calibrated density is now available, we have substituted the spacecraft potential by the actual density. This changes nothing in terms of the discussion and conclusions.

 – 'This the density is higher' – how much higher, and where? How is this visible in the data shown? Fig 2 caption – add comment (see definitions in text), or add a short explanation for the definition of the parameters shown

Previously this was not visible in the data, as we could not show the density. Since we have now included the density instead of the spacecraft potential this should be taken care off. Fig. 2 caption was expanded to be more self-explanatory.

 – 'transition can sometimes be very broad' – can this be quantified e.g. with respect to the electron, proton and heavy ion gyroradius? (see e.g. Coates et al. 1990)

I think it is not clear how we can relate this transition time to a scale. We do not know the velocity of the boundary and we dare not make assumptions.

 typo 'where'

Corrected

 – as well as Deca et al, there were earlier papers on momentum balance in the AMPTE releases and in comets (see Coates et al. 2015, and references therein, eg Coates et al, JGR 1986, Johnstone et al., Geophys. Monograph 38, 1985, Coates et al, Adv Space Res 1988))

The Deca et al paper gives a particularly good example of the deflection at a comet with details on the different behaviour of all particle species (solar wind electrons, ions, cometary electrons, ions). This kind of deflection is not explicitly addressed in models based on a fluid approach. We have added Coates et al. 2015 as a reference for observations of deflection at 67P.

 – 'flux of electrons does increase downstream' – might some of this be associated with spacecraft potential changes?

This is unlikely, we corrected for the spacecraft potential effect in the spectra shown here (see section 2.1). For the energies that we are looking at here (60eV and over) the spacecraft potential has very little effect on the measured energy spectra.

 - 'different for electrons and protons' – and heavy ions?

Indeed. This was added in the text.

 – Might shocklets (e.g. Omidi et al.), and/or upstream cavities, be relevant

Omidi et al. 1984 conducted one-dimensional hybrid simulations with the aim of modelling the spacecraft encounters with comets 1P/Halley and 21P/Giacobini-Zinner. They found that for oblique interaction (cone angle $55°$), shocklets form in a region of large amplitude wave activity. These shocklets convect downstream, where they break up due to dispersion, and new ones form further upstream. Thus, the process is repeated in a way that resembles shock reformation at planets (Balogh et al 2013). Although it is possible that shocklets form and shock reformation occurs also at comet 67P under certain conditions, it is not the cause of the observations reported here. The shock encounters shown in the paper do not display the repetitive transitions in a wave-dominated region that would be expected for the shocklets reported by Omidi et al. This discussion was added to the text at the end of section 4.

 – 10s of minutes – how might this compare to gyroperiods/radii?

The gyrofrequency of a proton in a 20nT field is $2s^{-1}$, for water ions it is $0.1s^{-1}$. Thus, we are well above the gyroperiod scales. In fact, all transition times that we observe are longer than the gyro period.

 – please specify/clarify/indicate on Fig 6 the times discussed (first/second half)

The term first and second half refer to the times with and without ICA observations. As the missing data is quite clear in the figure we do not think it is necessary to mark it, however, the text was rephrased to make it clear that we refer to the time with/without ICA spectra.

 – 'density of the plasma does not change significantly' – if anything, the spacecraft potential is more negative, thus density higher, in the 'upstream' region in this case

This statement seems to have been unclear. This sentence was supposed to refer to the average

behaviour, not the specific event shown in Figure 6. This was clarified in the text, and we have also pointed out that events with increased, decreased, and unchanged density can all be found.

Line 285 – could calculate the ratio between the solar wind and the local plasma density
Indeed, this is a good point. We cannot do this for all events due to data quality of the proton moments, but we did take a look at the event from Figure 1 and found a proton density of ca $0.5\,\mathrm{cm}^{-3}$ for this event. The plasma density is of the order of $1000\,\mathrm{cm}^{-3}$, this would mean a proton fraction of 0.05%. This seems rather low, however, the proton density estimate is also extremely low. Even assuming that ICA underestimate by a factor of 10, would only give us a fraction of 0.5%. Thus, for the larger plasma dynamics, the protons can be neglected. Alternatively we can make an estimate of the maximum proton density based on a simple fluid model, which seems a better way to get the maximum fraction of protons in the plasma. We have added to the paper: " We can estimate the fraction of cometary ions for the event shown in Fig. 1. The cometary ion density is of the order of $1000\,\mathrm{cm}^{-3}$ and we can estimate the maximum proton density from a simple back-of-the-envelope calculation: assuming a solar wind density of $3\,\mathrm{cm}^{-3}$ (typical for heliocentric distances around 2 AU) and a compression factor of $\sim 4$, we get a proton density of $12\,\mathrm{cm}^{-3}$. This is close to what is also observed in the simulation used below. This gives a fraction of $\sim 99\%$ cometary ions. Even if this estimate is very rough, it is clear that the cometary ions are at this point clearly dominating the plasma and the solar wind has only very little influence on the plasma density."

Line 288 – it would be useful to mention the assumed gas production rate Q in simulation and for the relevant observation
The gas production rate for the simulation was $3.2 \times 10^{27}\,\mathrm{s}^{-1}$. We added this to the text.

Line 290 – please indicate the suggested 'IBS' location on Fig. 5
A description was added in the caption.

Line 294 – what is the scale of proton gyration compared to the features seen in the simulation
The gyroradii of protons in the $200 - 400\,\mathrm{km\,s}^{-1}$ range are $100 - 200\,\mathrm{km}$ in a $20\,\mathrm{nT}$ magnetic field. This is comparable to the thickness of the infant bow shock. The typical length scale of the structure in the upper left corner of Fig. 5 is about $10^3\,\mathrm{km}$, corresponding to approximately 2 gyroradii in the weaker magnetic field ($\sim 10\,\mathrm{nT}$) in that region. This was added in the text.

Line 299 – Does +Ec correspond to Eparallelz as on the Figure?
Indeed, a clarification was added to the caption of the figure.

Line 306 – 'not significant enough to form a large bow shock' – rather than 'large' do you mean fully developed? Might there be a relation to shocklets?
On shocklets, see comment above. Yes, *fully developed* is a better descriptor, this was changed.

Line 322 – Re shock motion – as mentioned above, it should be possible to estimate the shock motion speed from the change in velocity and shock normal (e.g. Smith et al, Coates et al)
In general the normal is not well known, since knowing it would require measurements on both sides for the same conditions, and we generally only have slow transitions, where the conditions are likely to change.

Line 325 – please briefly explain the term 'caustic' Line 334 – re Comet Interceptor, depending on the gas production rate of the target comet, any observed cometary bow shock may be more fully developed than the features discussed here
Caustic is the term used in the referenced paper. A half-sentence describing it in more detail was added. Re Comet Interceptor, since the gas production rate of the comet to be visited is unknown, it is entirely possible to encounter an infant bow shock or a fully developed bow shock. We added an appropriate modifier in the text.

Line 340 – also, 3D fully kinetic simulations would be valuable
Indeed. This was clarified.

*Line 345 – refers to a 'density proxy' – is this the spacecraft potential? In Fig 6 the density appears higher upstream*
Since we now include the density, this is solved.

*Line 355 – More accurate to say 'It may be that the 'infant bow shock' is the low production rate manifestation of what becomes the more developed cometary bow shock as observed at larger comets such as Halley' (add references). Also discuss shocklets in this context*
This was reformulated. Shocklets are discussed in more detail in the previous section and since they are unrelated to the IBS we don't think a discussion of this belongs in the conclusion.

*Line 357 – 'ordinary' may not be the correct adjective for the complex bow shock structure, with changes at proton and heavy ion gyroscales, as observed at comets such as Halley (e.g. Coates et al., 1987).*
True. This was solved by adjusting the sentence above.

**Significant Text Changes**

[revised manuscript text omitted]

A. J. Coates, A. D. Johnstone, R. L. Kessel, D. E. Huddleston, and B. Wilken. Plasma parameters near the comet Halley bow shock. *J. Geophys. Res.*, 95:20701–20716, December 1990. doi: 10.1029/JA095iA12p20701.

A. J. Coates, A. D. Johnstone, and F. M. Neubauer. Cometary ion pressure anisotropies at comets Halley and Grigg-Skjellerup. *J. Geophys. Res.*, 101(A12):27573–27584, December 1996. doi: 10.1029/96JA02524.

A. J. Coates, C. Mazelle, and F. M. Neubauer. Bow shock analysis at comets Halley and Grigg-Skjellerup. *J. Geophys. Res.*, 102(A4):7105–7113, April 1997. doi: 10.1029/96JA04002.

C. Goetz, C. Koenders, K. C. Hansen, J. Burch, C. Carr, A. Eriksson, D. Frühauff, C. Güttler, P. Henri, H. Nilsson, I. Richter, M. Rubin, H. Sierks, B. Tsurutani, M. Volwerk, and K. H. Glassmeier. Structure and evolution of the diamagnetic cavity at comet 67P/Churyumov-Gerasimenko. *MNRAS*, 462:S459–S467, November 2016a. doi: 10.1093/mnras/stw3148.

C. Goetz, C. Koenders, I. Richter, K. Altwegg, J. Burch, C. Carr, E. Cupido, A. Eriksson, C. Güttler, P. Henri, P. Mokashi, Z. Nemeth, H. Nilsson, M. Rubin, H. Sierks, B. Tsurutani, C. Vallat, M. Volwerk, and K.-H. Glassmeier. First detection of a diamagnetic cavity at comet 67P/Churyumov-Gerasimenko. *A&A*, 588:A24, April 2016b. doi: 10.1051/0004-6361/201527728.

Charlotte Götz, Herber Gunell, Martin Volwerk, Arnaud Beth, Anders Eriksson, Marina Galand, Pierre Henri, Hans Nilsson, Cyril Simon Wedlund, Markku Alho, Laila Andersson, Nicolas Andre, Johan De Keyser, Jan Deca, Yasong Ge, Karl-Heinz Glaßmeier, Rajkumar Hajra, Tomas Karlsson, Satoshi Kasahara, Ivana Kolmasova, Kristie LLera, Hadi Madanian, Ingrid Mann, Christian Mazelle, Elias Odelstad, Ferdinand Plaschke, Martin Rubin, Beatriz Sanchez-Cano, Colin Snodgrass, and Erik Vigren. Cometary Plasma Science – A White Paper in response to the Voyage 2050 Call by the European Space Agency. *arXiv e-prints*, art. arXiv:1908.00377, Aug 2019.

Herbert Gunell, Charlotte Goetz, Cyril Simon Wedlund, Jesper Lindkvist, Maria Hamrin, Hans Nilsson, Kristie Llera, Anders Eriksson, and Mats Holmström. The infant bow shock: a new frontier at a weak activity comet. *A&A*, 619:L2, November 2018. doi: 10.1051/0004-6361/201834225.

F. L. Johansson, A. I. Eriksson, N. Gilet, P. Henri, G. Wattieaux, M. G. G. T. Taylor, C. Imhof, and F. Cipriani. A charging model for the Rosetta spacecraft. *A&A*, 642:A43, October 2020. doi: 10.1051/0004-6361/202038592.

R. L. Kessel, A. J. Coates, U. Motschmann, and F. M. Neubauer. Shock normal determination for multiple-ion shocks. *J. Geophys. Res.*, 99(A10):19359–19374, October 1994. doi: 10.1029/94JA01234.

C. Koenders, K.-H. Glassmeier, I. Richter, U. Motschmann, and M. Rubin. Revisiting cometary bow shock positions. *Planetary and Space Science*, 87:85–95, October 2013. doi: 10.1016/j.pss.2013.08.009.

K. E. Mandt, A. Eriksson, N. J. T. Edberg, C. Koenders, T. Broiles, S. A. Fuselier, P. Henri, Z. Nemeth, M. Alho, N. Biver, A. Beth, J. Burch, C. Carr, K. Chae, A. J. Coates, E. Cupido, M. Galand, K.-H. Glassmeier, C. Goetz, R. Goldstein, K. C. Hansen, J. Haiducek, E. Kallio, J.-P. Lebreton, A. Luspay-Kuti, P. Mokashi, H. Nilsson, A. Opitz, I. Richter, M. Samara, K. Szego, C.-Y. Tzou, M. Volwerk, C. Simon Wedlund, and G. Stenberg Wieser. RPC observation of the development and evolution of plasma interaction boundaries at 67P/Churyumov-Gerasimenko. *MNRAS*, 462:S9–S22, November 2016. doi: 10.1093/mnras/stw1736.

F. M. Neubauer, K. H. Glassmeier, M. Pohl, J. Raeder, M. H. Acuña, L. F. Burlaga, N. F. Ness, G. Musmann, F. Mariani, M. K. Wallis, E. Ungstrup, and H. U. Schmidt. First results from the Giotto magnetometer experiment at comet Halley. *Nature*, 321:352–355, May 1986. doi: 10.1038/321352a0.

H. Nilsson, G. Stenberg Wieser, E. Behar, H. Gunell, M. Galand, C. Simon Wedlund, M. Alho, C. Goetz, M. Yamauchi, P. Henri, and E. Odelstad A.I. Eriksson. Evolution of the ion environment of comet 67P during the rosetta mission as seen by RPC-ICA. *Monthly Notices of the Royal Astronomical Society*, 469 (Suppl_2):S252–S261, 2017. doi: 10.1093/mnras/stx1491.

N. Omidi and D. Winske. A kinetic study of solar wind mass loading and cometary bow shocks. *J. Geophys. Res.*, 92(A12):13409–13426, December 1987. doi: 10.1029/JA092iA12p13409.

C. Simon Wedlund, E. Behar, H. Nilsson, M. Alho, E. Kallio, H. Gunell, D. Bodewits, K. Heritier, M. Galand, A. Beth, M. Rubin, K. Altwegg, M. Volwerk, G. Gronoff, and R. Hoekstra. Solar wind charge exchange in cometary atmospheres III. Results from the Rosetta mission to comet 67P/Churyumov-Gerasimenko. *Astronomy & Astrophysics*, 2019. doi: 10.1051/0004-6361/201834881.

Edward J. Smith, Bruce T. Tsurutani, James A. Slavin, Douglas E. Jones, George L. Siscoe, and D. Asoka Mendis. International cometary explorer encounter with Giacobini-Zinner: Magnetic field observations. *Science*, 232(4748):382–385, 1986. ISSN 0036-8075. doi: 10.1126/science.232.4748.382.

**A   Additional events**

[Figure]

Figure 6: Observations of the plasma for the events shown in Table 1. Format is the same as in Figure 1.

[Figure]

Figure 7: Observations of the plasma for the events shown in Table 1. Format is the same as in Figure 1.

[Figure]

Figure 1: Left: Event from Figure 1. Right: Event from Figure 6.

---

## Author Comment (AC3) · 26 Feb 2021

**Response to Referee #3**
**Warm protons at comet 67P/Churyumov-Gerasimenko – Implications for the infant bow shock**

We thank the referee for the constructive comments and suggestions. We have made the necessary amendments to the paper and answers to comments may be found below. (Blue: Referee comment, black: our answer).

*Near Line 70, in this paper you are mainly exploring the characteristics in the data when the spacecraft crossed the infant shock. Can you also briefly mention and cite some references on what the data will be like if an ordinary or classical shock is crossed, so that readers can easily see the similarities and differences between the infant shock and the ordinary shock.*

It becomes clear now that the short introduction into the topic of classical bow shocks is not exensive enough. Therefore we added a more detailed summary of this topic.

*Line 159 & 160: "Interestingly, the flux diminishes at the same time that the proton energy increases gradually." Can you add some theoretical explanation to this phenomena?*

A detailed theoretical explanation will be quite difficult, but something similar was already observed by Gunell et al (2018). We think that the spacecraft is slowly transitioning into a different region behind the IBS, just as observed in the first event described by Gunell et al (2018). We have added a reference to this in the text.

*Line 163: the angle between the x-axis and magnetic field -¿ the angle between the x-axis and electric field?*

Both of these are correct, because we estimate the electric field from the magnetic field. The electric field is the more physically relevant parameter, so we have changed it in the text.

*Line 166: Does spacecraft attitude mean spacecraft orientation? Can you explain what are $\alpha_{x,y,z}$ of the spacecraft attitude?*

Yes, spacecraft attitude means the orientation of the spacecraft in a certain frame of reference. $\alpha_{x,y,z}$ are the angles of the spacecraft axes $(x, y, z)$ to the Comet-Sun line. However, we have since removed the spacecraft attitude from these figures as it provided little information. Instead we added in the Data Section that events with attitude changes above $> 10°$ were discarded.

*Line 173: "we find that the energy of the electrons is almost always increased". Is it consistent with your expectations? Can you add explanation the increase of electron energy and decrease in ion energy?*

It is indeed consistent with expectations. In this section we limit ourselves to the description of the data analysis, the discussion is done in the later section, where the text addresses this finding.

*Line 244: The statement "at least some of this discrepancy might be attributable to the inability of the flux at 60eV or 120eV to accurately represent the electron spectra" is not clear to me. Can you elaborate this point?*

The electron flux measured by IES depends on the FOV of the instrument as well as the spacecraft charge. Depending on these parameters, the measured flux will deviate significantly from the true state of the plasma, and this is of course also reflected in the flux at these specific two energies. A clarification that relates this back to the instrument caveats in section 2.1 was added.

*Line 2: after "infant bow shock" add "(IBS)".*
*Line 25: "and with it the amount of ice" -¿ "with increasing amount of ice"*
*Line 40: lower gyroradii -¿ smaller gyroradii*
*Line 49: "the comet's, frame of reference" -¿ "the comet's frame of reference"*
*Line 66: "insure" -¿ "ensure"*
*Line 74: "it's characteristics" -¿ "its characteristics"*
*Line 119: "instead" -¿ "because" ?*
*Line 138: by-eye inspection -¿ inspection by eyeball ?*

All these typos have been corrected.

**Significant Text Changes**

[revised manuscript text omitted]

A. J. Coates, A. D. Johnstone, R. L. Kessel, D. E. Huddleston, and B. Wilken. Plasma parameters near the comet Halley bow shock. *J. Geophys. Res.*, 95:20701–20716, December 1990. doi: 10.1029/JA095iA12p20701.

A. J. Coates, A. D. Johnstone, and F. M. Neubauer. Cometary ion pressure anisotropies at comets Halley and Grigg-Skjellerup. *J. Geophys. Res.*, 101(A12):27573–27584, December 1996. doi: 10.1029/96JA02524.

A. J. Coates, C. Mazelle, and F. M. Neubauer. Bow shock analysis at comets Halley and Grigg-Skjellerup. *J. Geophys. Res.*, 102(A4):7105–7113, April 1997. doi: 10.1029/96JA04002.

C. Goetz, C. Koenders, K. C. Hansen, J. Burch, C. Carr, A. Eriksson, D. Frühauff, C. Güttler, P. Henri, H. Nilsson, I. Richter, M. Rubin, H. Sierks, B. Tsurutani, M. Volwerk, and K. H. Glassmeier. Structure and evolution of the diamagnetic cavity at comet 67P/Churyumov-Gerasimenko. *MNRAS*, 462:S459–S467, November 2016a. doi: 10.1093/mnras/stw3148.

C. Goetz, C. Koenders, I. Richter, K. Altwegg, J. Burch, C. Carr, E. Cupido, A. Eriksson, C. Güttler, P. Henri, P. Mokashi, Z. Nemeth, H. Nilsson, M. Rubin, H. Sierks, B. Tsurutani, C. Vallat, M. Volwerk, and K.-H. Glassmeier. First detection of a diamagnetic cavity at comet 67P/Churyumov-Gerasimenko. *A&A*, 588:A24, April 2016b. doi: 10.1051/0004-6361/201527728.

Charlotte Götz, Herber Gunell, Martin Volwerk, Arnaud Beth, Anders Eriksson, Marina Galand, Pierre Henri, Hans Nilsson, Cyril Simon Wedlund, Markku Alho, Laila Andersson, Nicolas Andre, Johan De Keyser, Jan Deca, Yasong Ge, Karl-Heinz Glaßmeier, Rajkumar Hajra, Tomas Karlsson, Satoshi Kasahara, Ivana Kolmasova, Kristie LLera, Hadi Madanian, Ingrid Mann, Christian Mazelle, Elias Odelstad, Ferdinand Plaschke, Martin Rubin, Beatriz Sanchez-Cano, Colin Snodgrass, and Erik Vigren. Cometary Plasma Science – A White Paper in response to the Voyage 2050 Call by the European Space Agency. *arXiv e-prints*, art. arXiv:1908.00377, Aug 2019.

Herbert Gunell, Charlotte Goetz, Cyril Simon Wedlund, Jesper Lindkvist, Maria Hamrin, Hans Nilsson, Kristie Llera, Anders Eriksson, and Mats Holmström. The infant bow shock: a new frontier at a weak activity comet. *A&A*, 619:L2, November 2018. doi: 10.1051/0004-6361/201834225.

F. L. Johansson, A. I. Eriksson, N. Gilet, P. Henri, G. Wattieaux, M. G. G. T. Taylor, C. Imhof, and F. Cipriani. A charging model for the Rosetta spacecraft. *A&A*, 642:A43, October 2020. doi: 10.1051/0004-6361/202038592.

R. L. Kessel, A. J. Coates, U. Motschmann, and F. M. Neubauer. Shock normal determination for multiple-ion shocks. *J. Geophys. Res.*, 99(A10):19359–19374, October 1994. doi: 10.1029/94JA01234.

C. Koenders, K.-H. Glassmeier, I. Richter, U. Motschmann, and M. Rubin. Revisiting cometary bow shock positions. *Planetary and Space Science*, 87:85–95, October 2013. doi: 10.1016/j.pss.2013.08.009.

K. E. Mandt, A. Eriksson, N. J. T. Edberg, C. Koenders, T. Broiles, S. A. Fuselier, P. Henri, Z. Nemeth, M. Alho, N. Biver, A. Beth, J. Burch, C. Carr, K. Chae, A. J. Coates, E. Cupido, M. Galand, K.-H. Glassmeier, C. Goetz, R. Goldstein, K. C. Hansen, J. Haiducek, E. Kallio, J.-P. Lebreton, A. Luspay-Kuti, P. Mokashi, H. Nilsson, A. Opitz, I. Richter, M. Samara, K. Szego, C.-Y. Tzou, M. Volwerk, C. Simon Wedlund, and G. Stenberg Wieser. RPC observation of the development and evolution of plasma interaction boundaries at 67P/Churyumov-Gerasimenko. *MNRAS*, 462:S9–S22, November 2016. doi: 10.1093/mnras/stw1736.

F. M. Neubauer, K. H. Glassmeier, M. Pohl, J. Raeder, M. H. Acuña, L. F. Burlaga, N. F. Ness, G. Musmann, F. Mariani, M. K. Wallis, E. Ungstrup, and H. U. Schmidt. First results from the Giotto magnetometer experiment at comet Halley. *Nature*, 321:352–355, May 1986. doi: 10.1038/321352a0.

H. Nilsson, G. Stenberg Wieser, E. Behar, H. Gunell, M. Galand, C. Simon Wedlund, M. Alho, C. Goetz, M. Yamauchi, P. Henri, and E. Odelstad A.I. Eriksson. Evolution of the ion environment of comet 67P during the rosetta mission as seen by RPC-ICA. *Monthly Notices of the Royal Astronomical Society*, 469 (Suppl_2):S252–S261, 2017. doi: 10.1093/mnras/stx1491.

N. Omidi and D. Winske. A kinetic study of solar wind mass loading and cometary bow shocks. *J. Geophys. Res.*, 92(A12):13409–13426, December 1987. doi: 10.1029/JA092iA12p13409.

C. Simon Wedlund, E. Behar, H. Nilsson, M. Alho, E. Kallio, H. Gunell, D. Bodewits, K. Heritier, M. Galand, A. Beth, M. Rubin, K. Altwegg, M. Volwerk, G. Gronoff, and R. Hoekstra. Solar wind charge exchange in cometary atmospheres III. Results from the Rosetta mission to comet 67P/Churyumov-Gerasimenko. *Astronomy & Astrophysics*, 2019. doi: 10.1051/0004-6361/201834881.

Edward J. Smith, Bruce T. Tsurutani, James A. Slavin, Douglas E. Jones, George L. Siscoe, and D. Asoka Mendis. International cometary explorer encounter with Giacobini-Zinner: Magnetic field observations. *Science*, 232(4748):382–385, 1986. ISSN 0036-8075. doi: 10.1126/science.232.4748.382.

**A    Additional events**

[Figure]

Figure 6: Observations of the plasma for the events shown in Table 1. Format is the same as in Figure 1.

[Figure]

Figure 7: Observations of the plasma for the events shown in Table 1. Format is the same as in Figure 1.